# Self-Guiding Exploration for Combinatorial Problems

**Zangir Iklassov**
MBZUAI
zangir.iklassov@mbzuai.ac.ae

**Yali Du**
King's College London
yali.du@kcl.ac.uk

**Farkhad Akimov**
MBZUAI
farkhad.akimov@mbzuai.ac.ae

**Martin Takáč**
MBZUAI
martin.takac@mbzuai.ac.ae

## Abstract

Large Language Models (LLMs) have become pivotal in addressing reasoning tasks across diverse domains, including arithmetic, commonsense, and symbolic reasoning. They utilize prompting techniques such as Exploration-of-Thought, Decomposition, and Refinement to effectively navigate and solve intricate tasks. Despite these advancements, the application of LLMs to Combinatorial Problems (CPs), known for their NP-hardness and critical roles in logistics and resource management remains underexplored. To address this gap, we introduce a novel prompting strategy: Self-Guiding Exploration (SGE), designed to enhance the performance of solving CPs. SGE operates autonomously, generating multiple thought trajectories for each CP task. It then breaks these trajectories down into actionable subtasks, executes them sequentially, and refines the results to ensure optimal outcomes. We present our research as the first to apply LLMs to a broad range of CPs and demonstrate that SGE outperforms existing prompting strategies by over 27.84% in CP optimization performance. Additionally, SGE achieves a 2.46% higher accuracy over the best existing results in other reasoning tasks (arithmetic, commonsense, and symbolic).

## 1 Introduction

Large Language Models (LLMs) have emerged as powerful tools capable of executing reasoning tasks across various domains, including arithmetic, commonsense, and symbolic reasoning [4, 33, 5, 27]. These models may leverage prompting techniques such as Exploration-of-Thought [39, 13, 42], Decomposition [49, 12], and Refinement [19] to break down and solve various tasks in a step-by-step manner. Recent research has been directed towards extending these techniques to tackle more sophisticated optimization challenges [41]. Combinatorial problems (CPs) may represent a category of these complex optimization tasks, associated with intricate computational challenges.

Combinatorial Problems are characterized by their NP-hardness and inherent complexity, which result in an exponential growth in the number of potential solutions. This complexity presents substantial challenges in the research [26, 25, 11, 10]. CPs are especially crucial in sectors that require efficient logistics, planning, and scheduling. Currently, the dominant approach in these industries involves metaheuristic methods. These methods combine various simple but fast heuristics to effectively tackle CPs within specific constraints. Nonetheless, the effectiveness of these heuristics can vary significantly depending on the CP task and its associated constraints, necessitating a customized selection of heuristics to achieve optimal performance.

38th Conference on Neural Information Processing Systems (NeurIPS 2024).

In the meantime, research on exploring LLMs to solve CPs reveals substantial gaps. While recent advancements indicate the effectiveness of LLMs in various reasoning tasks [38, 47, 32, 49], their application to CPs has been minimal. The literature indicates that existing generative models can address smaller instances of the Traveling Salesman Problem (TSP) [16, 23, 41]. However, as problem sizes increase, existing prompting strategies begin to yield inadequate responses, underscoring the need for more sophisticated prompting methods. Moreover, there is a notable scarcity of research addressing other complex CPs, particularly the Vehicle Routing and Job Scheduling Problems, which pose significant challenges in logistics, planning industries, and operations research.

In this work, we introduce a novel prompting strategy: self-guiding exploration (SGE), designed to enhance the problem-solving process for CPs. This algorithm works as a combination of exploration-of-thought, decomposition, and refinement prompting methods. The SGE approach autonomously generates multiple thought trajectories for a given CP, each trajectory representing a specific heuristic to tackle the given task. Each trajectory is then decomposed into subtasks, which are executed one by one, and their outputs are refined and combined into a final solution. Unlike the task-specific prompts utilized in other methods, SGE employs general-purpose prompts, allowing for the adaptive use of specific heuristic solutions tailored to various CPs, such as the Hungarian heuristic for the assignment problem and the Nearest Neighbor heuristic for the vehicle routing problem. Essentially, SGE acts as a versatile metaheuristic capable of identifying, combining, and refining task-specific heuristics for individual CP tasks.

**Our work makes following contributions.** Firstly, we present a novel investigation into the application of large language models for solving combinatorial problems. Secondly, we introduce a new prompting strategy, SGE, that autonomosly generates thought trajectories, splits them into subtasks and refines the answers. Thirdly, we demonstrate that SGE outperforms existing prompting strategies such as Chain-of-Thought, Decomposition, and Self-Refinement, improving CP optimization performance by $27.84\%$. Lastly, we validate the applicability of SGE across other reasoning tasks, including arithmetic, commonsense, and symbolic tasks, where our method achieves a $2.46\%$ higher accuracy than the best existing results.

## 2 Related work

**CP via classical approach.** In classical research, combinatorial problems are predominantly tackled using heuristic and metaheuristic methods specifically crafted for particular tasks. Notable examples include the Ant-Colony Optimization and Tabu Search methods for addressing the Vehicle Routing Problem [28, 8, 15], the Shortest Processing Time and Most Work Remaining Heuristics for Job Scheduling Problems [30], and the Hungarian Algorithm for the Assignment Problem [1]. These approaches are favored in industrial settings due to their simplicity and speed. However, they need to be individually tailored to each task and its constraints' setting. In contrast, exact solvers such as Google-OR-Tools [9] offer general and precise solutions, but their applicability is often limited to smaller-scale problems due to the inherent NP-hardness of combinatorial problems.

**CP via learning-based approach.** In AI literature, Reinforcement Learning (RL) has been a prominent approach for tackling combinatorial problems since the 1990s [21, 20, 45, 7, 2]. The integration of deep learning, particularly through innovations like Pointer Networks, has significantly enhanced RL's capability to handle more complex combinatorial tasks [35, 3]. Further advancements involve the use of Transformer networks [6, 34, 14], with notable applications in solving the Vehicle Routing Problem [25, 11]. Despite these advances, RL-based methods often still do not exceed the performance of traditional heuristics, especially when scalability and accurate state representation are required [36, 22, 44, 31].

**CP with LLMs.** Recent studies have leveraged large language models (LLMs), such as GPT-3.5 and GPT-4, to tackle combinatorial problems like the Traveling Salesman Problem using iterative prompting, where solutions are refined incrementally [23, 41]. Other works employ LLMs to autonomously generate executable code as novel heuristics for problems like the knapsack and traveling salesman [29, 17, 43, 18]. This promising approach enhances task-specific heuristics, potentially improving performance on specialized combinatorial tasks. In contrast, our focus is on leveraging LLMs to directly solve combinatorial problems in a generalizable manner, enabling a versatile approach applicable across a wide range of complex tasks.

**Prompting strategies.**    The expressive capabilities of direct prompting in Large Language Models are theoretically limited to the complexity class $\mathsf{TC}^0$ [24]. To effectively address combinatorial problems with LLMs, sophisticated prompting strategies are required. One basic approach is the Chain-of-Thought (CoT) prompting, introduced in [40], which encourages LLMs to articulate intermediate "thoughts" that inform the generation of the final output. This technique has given rise to advanced variations, including Self-consistency with CoT (CoT-SC), Tree-of-Thoughts (ToT), and Graph-of-Thought methods [37, 42, 46]. Additionally, decomposition prompting strategies can be employed [48, 12], since they simplify complex tasks into smaller, manageable subtasks via symbolic programs or structured algorithms, thus improving the performance of LLMs. In our experiments, we found these techniques to be insufficient, leading us to propose the Self-Guiding Exploration method as a more effective solution for tackling combinatorial problem tasks.

## 3   Preliminaries

We provide an overview of combinatorial problems, highlighting their inherent complexity with the classic example of the Traveling Salesman Problem (TSP) and an example of a combinatorial problem formulation in a prompt for use by a Large Language Model (LLM).

**Combinatorial problems.**    Combinatorial problems involve decision-making processes where the goal is to assign binary decision variables $x \in 0, 1$ in order to optimize a cost function $g(x_1, ..., x_n)$, subject to task-specific constraints. A classic example of such a problem is the TSP. In the TSP, given a list of $n$ cities and the distances $d_{ij}$ between cities $i, j$, the objective is to determine the shortest possible route that visits each city exactly once and returns to the starting city. $x_{ij}$ is used as the action variable, indicating whether the route progresses from city $i$ to city $j$. The cost function to minimize in TSP is $g(x) = \sum_{i=1}^{n} \sum_{j=1}^{n} d_{ij} x_{ij}$, under the condition that all cities visited exactly once $\sum_{i=1}^{n} x_{ij} = 1$ and $\sum_{j=1}^{n} x_{ij} = 1$ for all $i, j$. Combinatorial problems are generally categorized as NP-hard due to their inherent computational complexity. For instance, a TSP with $n$ cities presents $(n-1)!$ possible routes, rendering the evaluation of all potential solutions impractical and exceedingly time-consuming as $n$ increases.

**Prompting combinatorial problems in LLMs.**    To use LLM for solving CP tasks, we define $f$ as the interface function of a generative LLM model, which accepts high-dimensional discrete input tokens and generates outputs within the same token space ($f : W \mapsto W$). For each combinatorial problem task, the input $Q$ to the LLM can be explicitly defined in a textual format. This description delineates the specific goal alongside a list of variables tailored to the task at hand. For instance, the objective of the Traveling Salesman Problem (TSP) can be textually articulated as "Find a route that minimizes the total travel distance, visits each city exactly once, and starts and ends in the same city." Subsequently, the variables, such as the distances between cities $d_{ij}$, are provided in a format such as "The distance between city $i$ and city $j$ is [number]", laying out all necessary parameters for the model to process and generate solutions. The model will then process this structured input $Q$ to produce the corresponding solution answer $A$, where both $Q$ and $A$ are within token space $W$; formally, $A = f(Q)$. Given the inherent complexity of combinatorial problems, direct zero-shot prompting $f(Q)$ is insufficient. Consequently, we propose a self-guiding exploration algorithm that employs metaheuristic-like strategies to effectively solve CP tasks.

## 4   Method

In this section, we introduce Self-Guiding Exploration (SGE) method and provide a detailed explanation of its algorithm designed to tackle combinatorial problems. The method (Fig 1), inspired by metaheuristic approaches, synthesizes multiple heuristic methods. It generates various thought trajectories, with each trajectory representing a specific heuristic approach. These trajectories are then integrated to form the final solution. To overcome the challenges of executing complex heuristics through LLMs in one step, our algorithm utilizes a decomposition strategy. This approach breaks down each trajectory into smaller, more manageable subtasks, enabling the solution to progress through sequential, simpler steps. This general-purpose algorithm is tailored to adapt to a wide range of combinatorial problems without the constraints of task-specific exemplars for few-shot solution generation.

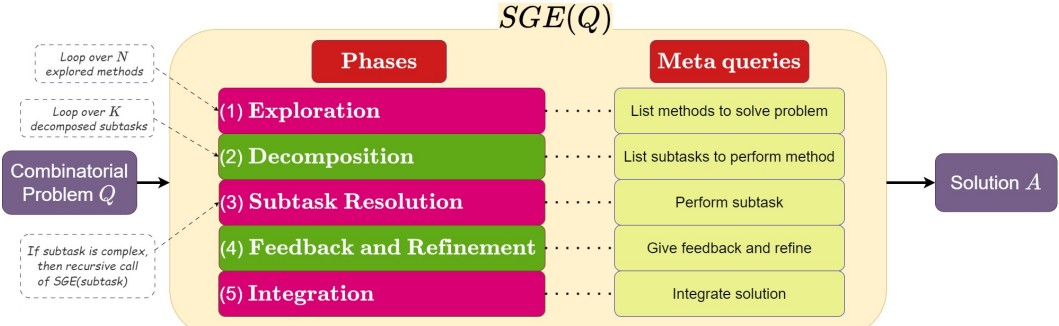

Figure 1: **Self-Guiding Exploration**. The generative model autonomously addresses a combinatorial problem task $Q$ through a five-phase process: (1) Exploration of $N$ solution trajectories, where each trajectory offers potential solutions; (2) Decomposition of these trajectories into $K$ subtasks, outlining specific steps for each method; (3) Resolution of each subtask, executing the outlined steps; (4) Feedback and Refinement, where feedback is gathered and used to refine each subtask; (5) Integration of all trajectories into a consolidated final solution $A$. Distinct from traditional exploration/decomposition techniques, SGE(Q) functions entirely autonomously, eliminating the reliance on task-specific queries or manually created thought exemplars. This independence makes it universally applicable to all CP tasks without necessitating modifications.

---

**Algorithm 1** Self-Guiding Exploration algorithm - $SGE(\cdot)$

---

**Require:** query $Q$, model $f$, meta-prompts $Z$, maximum recursion depth $D$

1:  $Q^{\mathbb{N}} = f(Q, Z_{explore})$                                              ▷ Explore method trajectories
2:  **for** iteration n $\in 1, 2, \ldots, N$ **do**
3:      $Q^n_{\mathbb{K}} = f(Q, Q^n, Z_{decomp})$                               ▷ Decompose trajectory subtasks
4:      **for** iteration k $\in 1, 2, \ldots, K$ **do**
5:          **if** $f(Q^n_k, Z_{check})$ **then**                          ▷ Check if subtask is simple
6:             $T^n_k = f(Q, T^n_{k-1}, Q^n_k)$               ▷ Execute subtask and get thought
7:          **else**
8:             $T^n_k = SGE(T^n_{k-1} || Q^n_k, f, Z)$            ▷ Recursive call of subtask
9:          **end if**
10:        $Q^n_{k_{feedback}} = f(Q, Q^n_k, T^n_k, Z_{feedback})$          ▷ Get feedback query
11:        $T^n_k = f(Q, T^n_k, Q^n_{k_{feedback}})$                     ▷ Refine thought
12:      **end for**
13: **end for**
14: $A = f(Q, T^1_K, \ldots, T^N_K, Z_{integrate})$                         ▷ Get answer

---

### 4.1 Algorithm

The proposed method's algorithm is segmented into five distinct phases, as outlined in Algorithm 1. These phases include exploring thought trajectories, decomposing each trajectory into subtasks, resolving each subtask to generate thoughts, obtaining feedback and refining the thoughts, and finally integrating all thoughts to formulate the answer. Each thought here represents one completed subtask.

**Exploration.** During the exploration stage, the model tackles the overarching problem $Q$ by engaging with exploration meta-prompt. This prompt $Z_{explore}$ is structured as: "List all possible methods to solve this problem. Return them separated by new lines." This prompt stimulates the model to enumerate potential methodologies pertinent to $Q$. The sequence is then divided into task-specific trajectories of queries $Q^n$ each incorporating a method to address $Q$:

$$Q^{\mathbb{N}} = f(Q, Z_{explore}).$$

**Decomposition.** Following exploration, each trajectory query $Q^n$ is processed through the model to break down the trajectory into actionable steps. The decomposition meta-prompt $Z_{decomp}$ is formulated as: "List all steps to use the method. Return them separated by new lines." This leads to

the generation of subtask queries that operationalize the trajectory method:

$$Q_{\mathbb{K}}^n = f(Q, Q^n, Z_{decomp}).$$

**Subtask resolution.**    Post-decomposition, the subtask queries $Q_{\mathbb{K}}^n$ are each split into $K$ individual queries and processed by the model to generate thoughts. The model initially evaluates if the task is easily solvable using the meta-prompt $Z_{check}$: "Is this problem easily solvable? Return yes or no": $f(Q_k^n, Z_{check})$. If the response is affirmative, the model executes the subtask query $Q_k^n$ to generate a new thought:

$$T_k^n = f(Q, T_{k-1}^n, Q_k^n).$$

Otherwise, the model engages a recursive instance of the self-guiding exploration algorithm on $Q_k^n$ instead of the main task $Q$ to navigate and decompose the complex subtask, producing:

$$T_k^n = f(Q, T_k^n, Q_{k_{feedback}}^n).$$

**Feedback and refinement.**    In this stage, the model utilizes an additional meta-prompt $Z_{feedback}$ - "Give feedback to the proposed solution" - to generate feedback queries $Q_{k_{feedback}}^n = f(Q, Q_k^n, T_k^n, Z_{feedback})$. This guides the model in refining the initial responses through reevaluation and enhancement of the thoughts:

$$T_k^n = f(Q, T_k^n, Q_{k_{feedback}}^n).$$

**Integration.**    Upon completion of all trajectories and their associated subtasks, the model employs a final meta-prompt $Z_{integrate}$ - "Integrate all previous findings and provide the final answer" - to amalgamate the last thoughts into a definitive solution answer:

$$A = f(Q, T_K^1, ...T_K^N, Z_{integrate}).$$

SGE draws inspiration from metaheuristic methods used to solve combinatorial problem tasks. Yet, due to its general-purpose nature and meta-prompts, it is also suitable for other tasks, beyond CPs. Essentially, it integrates elements of exploration-of-thought, decomposition, and refinement prompting strategies, but it does so without relying on task-specific prompts or solution exemplars. For additional information on these prompting strategies, see Section A.1

## 5    Experiments

This section details the experimental setup and presents the results of our proposed method applied to combinatorial problem tasks, as well as its performance on other reasoning tasks commonly explored in LLM research.

### 5.1    Setup

**CP tasks.**    The experiments were conducted on six combinatorial tasks: Assignment Problem, Knapsack Problem, Bin Packing Problem, Traveling Salesman Problem, Vehicle Routing Problem, and Job Scheduling Problem. The Assignment Problem, classified as P-complete, can be optimally solved using the Hungarian Algorithm. In contrast, the other tasks are NP-hard and ordered by increasing complexity. For a more detailed discussion of these CP tasks, refer to Section A.2. We included five distinct problem sizes, involving 5, 10, 15, 20, and 30 elements (nodes) such as cities in the TSP/VRP. To facilitate these experiments, a dataset was created, comprising 100 randomly generated instances for each problem size. These instances were characterized by uniformly distributed variables, such as the positioning of cities in TSP/VRP or bin volume in the Bin Packing Problem, over an interval from 0 to 100. The experiments utilized an NVIDIA A100 SXM 40GB GPU, paired with two AMD EPYC 7742 CPUs (8 cores each) and 256GB RAM. Our implementation is available online.[1]

---
[1]`https://github.com/Zangir/LLM-for-CP`

Table 1: **Percentage performance improvement compared to IO** on CP tasks using GPT-4 and Gemini-1.5 models. CoT uses majority voting, with the number of candidates equal to the number of thoughts produced by SGE. The metrics is quantified as percentage improvement in cost with respect to IO solution (the bigger it is the better).

| Task | GPT-4 | | | | Gemini-1.5 | | | |
|------|-------|--------|--------|-------|-------|--------|--------|-------|
| | CoT | Refine | Decomp | Ours | CoT | Refine | Decomp | Ours |
| Assignment | 11.46 | 14.47 | 33.80 | **41.33** | 11.66 | 13.98 | 31.94 | **40.46** |
| Knapsack | 15.37 | 17.16 | 51.95 | **70.39** | 13.85 | 16.85 | 48.62 | **65.87** |
| Bin Packing | 14.06 | 17.12 | 39.57 | **74.72** | 11.89 | 15.43 | 35.74 | **67.63** |
| Travelling Salesman | 13.64 | 15.75 | 38.49 | **72.10** | 14.34 | 15.90 | 36.36 | **68.09** |
| Vehicle Routing | 14.27 | 16.94 | 36.73 | **71.92** | 11.88 | 15.13 | 33.59 | **68.02** |
| Job Scheduling | 13.84 | 16.37 | 38.20 | **75.33** | 13.41 | 15.75 | 36.36 | **67.89** |

**Baselines.** We utilized four baseline prompting methods: Input-Output (IO) Direct Prompting, Chain-of-Thought Prompting, Self-Refine (Refine) Prompting, and Decomposition Prompting. The Input-Output (IO) approach involves a single prompt where the model is asked to provide a solution directly, without complex prompting. In this approach, we generate $N$ sample candidates by repeatedly prompting the model with the same query $Q$, $N$ times. The responses are then aggregated through majority voting to identify the most common solution among the $N$ outputs. We employ the Self-Refine (Refine) method [19], which includes a feedback-refinement procedure that aligns closely with phase four of SGE. Additionally, we use the zero-shot Chain-of-Thought method [13], which is the basic technique among Exploration-of-Thought methods. Lastly, we implemented the Decomposition method as described in [49]. In our experiments, these baseline methods were tested across a range of five LLM models including GPT-4, GPT-3.5 by OpenAI, Gemini-1.5 by Google, and the Llama-2 series from Meta, which includes models with 70 billion and 7 billion parameters. We did not include prompting methods previously used in [16, 23, 41], as their prompting strategies showed inferior results compared to the zero-shot Chain-of-Thought approach when tested with our data.

**Metrics.** In our study, each method's performance is evaluated relative to IO (Input-Output) prompting. To quantify the improvement, we first measure the solution cost $g_{io}$ for each combinatorial problem task using IO prompting (e.g., for the TSP, $g_{io} = \sum_{i=1}^{n} \sum_{j=1}^{n} d_{ij} x_{ij}$). We then calculate the cost $g_{method}$ using alternative methods. The percentage improvement is computed as $100 \times \frac{g_{io} - g_{method}}{g_{io}}$. For problems of smaller sizes, we are able to obtain optimal solutions using the Google-OR-Tools solver through a brute force approach. In such instances, we measure the cost of the optimal solution $g_{opt}$ and determine the optimality gap as $100 \times \frac{g_{method} - g_{opt}}{g_{opt}}$.

## 5.2 Results on CP tasks

To evaluate the general performance of SGE on combinatorial problems, we conducted experiments comparing performance improvement to IO of SGE and CoT, Decomposition, and Refinement baselines using GPT-4 and Gemini-1.5 LLM models. Table 1 gives the results on six combinatorial problem tasks. The results analysis shows that the SGE method consistently outperforms CoT, Refine, and Decomposition methods across all tasks. Notably, the magnitude of improvement escalates with the increasing complexity of the problems, from polynomial to exponential. The margin with the second-best method, Decomposition, ranges from 7.53% for the Assignment Problem to 37.13% for the JSP. This trend suggests that the IO method may struggle with the computational demands of NP-hard problems, where more sophisticated strategies like SGE provide significant advantages. For a comprehensive view of all experimental results, see Section A.4.

To qualitatively evaluate the performance of the Exploration, Decomposition, and Refinement phases of SGE method, we assessed the LLM outputs for each phase across five random instances of each combinatorial problem task. Figure 2 illustrates an example of how our method addresses the TSP, showcasing outputs during each of the three phases of SGE. In the Exploration phase, the first box of Figure 2 displays how LLM $f$ generates a list of potential algorithms suitable for solving

Figure 2: **Example of SGE inference** across the Exploration, Decomposition, and Refinement phases for the Traveling Salesman Problem. The figure displays three boxes, each illustrating the prompt structure and corresponding example output for each phase.

the TSP, such as heuristic approaches like Nearest Neighbor, metaheuristic techniques like Ant Colony, and Mixed Integer Linear Programming (MILP) method. This phase adapts to different combinatorial problems by suggesting tailored algorithms, like the Hungarian algorithm for the Assignment problem, Greedy algorithms for the Knapsack problem, and Clustering methods for the Vehicle Routing Problem (VRP). Each list of algorithms forms the foundation for generating diverse candidate solutions tailored to each specific problem. The Decomposition phase, depicted in the second box of Figure 2, breaks down each identified algorithm into specific subtasks. This example shows the decomposition of the Nearest Neighbor algorithm for the TSP, where the initial subtasks are simple enough for direct processing by model $f$. However, more complex tasks, such as loops, undergo further decomposition using SGE in a recursive manner, with computational or programming tasks being handled using Python within models like GPT-4 and Gemini-1.5 equipped with a Code Interpreter. Finally, the Refinement phase, illustrated in the third box of Figure 2, focuses on enhancing the candidate solutions developed in the previous stage. This example pertains to refining a solution derived from the Nearest Neighbor algorithm for the TSP by implementing the 2-opt algorithm. Renowned for its effectiveness in TSP and VRP contexts, the 2-opt algorithm optimizes the initial solution to find locally optimal solutions within a specific neighborhood, thus improving the overall quality of the candidate solutions. This example shows that SGE method adapts its approach to suit different combinatorial problems, finding a special set of heuristics for each task.

**Effect of problem size on SGE performance.**    To evaluate the effect of problem size on SGE performance, we conducted experiments on all six tasks with input sizes of 5, 8, 12, 15, and 20 nodes using the GPT-4 model. Figure 3 gives the results of these experiments. The results analysis shows that generally, an increase in problem complexity, as determined by the size of the problem input, negatively influences performance improvement; larger problem sizes result in diminished performance improvement of the SGE method compared to IO. Specifically, tasks with 20 input nodes consistently exhibit lower performance improvements relative to the IO method than tasks with 5 input nodes. However, when comparing less disparate sizes, such as 8 and 12 nodes, the differential impact on performance is less pronounced and can occasionally be positive.

Table 2: **Optimality gap** of prompting methods using `LLaMA-2-70B`. The results are represented as performance percentage difference compared to optimal solutions (the smaller it is, the better).

| Size | Method | Assignment | Knapsack | Bin Packing | TSP | VRP | JSP |
|---|---|---|---|---|---|---|---|
| 5 NODES | IO | 45.45 | 90.10 | 108.2 | 100.3 | 102.0 | 105.3 |
| | CoT | 39.33 | 66.88 | 78.24 | 81.15 | 78.17 | 79.41 |
| | Refine | 36.42 | 61.98 | 77.40 | 71.62 | 72.49 | 71.72 |
| | Decomp | 14.66 | 21.56 | 40.00 | 43.62 | 40.65 | 44.15 |
| | Ours | **2.500** | **8.050** | **9.060** | **8.27** | **11.92** | **9.300** |
| 8 NODES | IO | 46.84 | 103.5 | 112.8 | 116.9 | 116.3 | 108.2 |
| | CoT | 39.70 | 73.84 | 85.08 | 89.01 | 89.48 | 85.21 |
| | Refine | 37.32 | 72.62 | 86.25 | 85.59 | 83.31 | 78.43 |
| | Decomp | 18.49 | 26.43 | 52.73 | 53.48 | 54.43 | 49.81 |
| | Ours | **8.290** | **14.88** | **20.95** | **15.19** | **19.65** | **21.26** |
| 12 NODES | IO | 49.11 | 101.5 | 120.7 | 121.6 | 118.5 | 117.6 |
| | CoT | 41.70 | 79.33 | 93.84 | 86.84 | 90.05 | 89.29 |
| | Refine | 40.35 | 77.09 | 82.23 | 88.57 | 88.40 | 87.02 |
| | Decomp | 21.12 | 35.82 | 55.40 | 57.51 | 59.19 | 56.01 |
| | Ours | **11.26** | **16.82** | **22.38** | **16.12** | **24.00** | **22.86** |

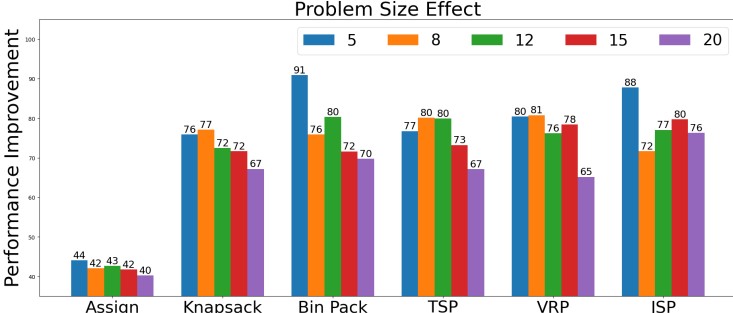

Figure 3: **Effect of Problem Size on Performance Improvement** relative to the IO solution using `gpt-4` w/ code interpreter. The analysis spans problem instances of varying sizes, systematically presented from the smallest to the largest, specifically ranging from $n = 5$ to $n = 20$ nodes. Results are organized to highlight the impact of increasing problem complexity on the effectiveness of the solution.

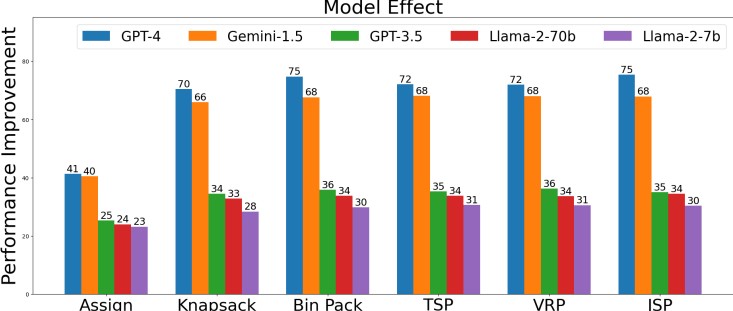

Figure 4: **Effect of Model Choice on Performance Improvement** relative to the IO solution.

Table 3: **Results for reasoning tasks** using `gpt-4` with a code interpreter are presented as accuracies on benchmark test sets.

| Method | Arithmetic | | | | Commonsense | | | Symbolic | Avg. |
|---|---|---|---|---|---|---|---|---|---|
| | AQUA | GSM8K | SVAMP | ASDiv | StrategyQA | CSQA | ARC | LastLetter | |
| IO Prompting | 67.30 | 87.04 | 88.34 | 90.10 | 78.40 | 81.14 | 87.52 | 81.98 | 82.73 |
| CoT Prompting | 69.57 | 89.76 | 91.58 | 93.32 | 81.16 | 84.15 | 90.80 | 85.18 | 85.69 |
| Refine Prompting | 69.68 | 89.80 | 91.00 | 93.10 | 81.26 | 83.50 | 91.09 | 84.82 | 85.53 |
| Decomp Prompting | 69.99 | 91.85 | 92.16 | 94.08 | 82.08 | 84.88 | 91.76 | 85.64 | 86.43 |
| Ours | **74.63** | **97.35** | **98.16** | **97.24** | **83.49** | **85.68** | **93.28** | **86.96** | **89.60** |

Table 4: **Performance vs efficiency** of prompting methods using `gpt-4` w/ code interpreter. The results are represented as performance percentage improvement compared to IO solution and number of model $f$ calls.

| Method | Performance Improvement | Function Calls |
|---|---|---|
| CoT | 13.77 | 58.32 |
| Refine | 16.30 | 58.32 |
| Decomp | 39.79 | 31.04 |
| Ours | 67.63 | 58.32 |

**Gap between SGE performance and the global optimum.** To evaluate the gap between SGE solutions and global optimum solutions, we conducted experiments on small problem sizes involving 5, 8, and 12 nodes, utilizing the brute force method via Google-OR-Tools to determine the global optimum. Figure 2 provides the results of these experiments in terms of the percentage gap between the performance of SGE and optimal solutions. The results analysis shows that generally, the SGE method exhibits a smaller optimality gap across all tasks when compared to baseline methods. This advantage is particularly pronounced for more complex problems such as Bin Packing, TSP, VRP, and JSP. For instance, SGE achieves a 12.16% smaller optimality gap on the Assignment task than the next best Decomposition method and a 34.85% smaller gap on the JSP.

**Effect of LLM selection on SGE performance.** To evaluate the impact of model selection on SGE performance, we conducted experiments comparing different models, including GPT-4, Gemini-1.5, GPT-3.5, Llama-2-70b, and Llama-2-7b models. Figure 4 provides the results of these experiments. The results analysis shows that GPT-4 and Gemini-1.5 demonstrate significantly better performance compared to other models across all tasks. A notable feature of both models is the integration of a Code Interpreter (CI) tool, which appears crucial for combinatorial problem tasks as it enables the models to execute generated code and evaluate solution performance directly. In contrast, models lacking a CI tool, such as GPT-3.5, Llama-2-70b, and Llama-2-7b, exhibit poorer outcomes, with GPT-3.5 slightly outperforming the Llama models. The comparison between Llama models indicates that the size of the model with 70 billion versus 7 billion parameters does not significantly influence performance. This suggests that model size alone does not guarantee substantial performance improvements in combinatorial tasks.

**Trade-off between performance and cost in SGE.** To evaluate the cost-effectiveness of different methods, we conducted experiments to compare the average performance improvement per method against the average number of LLM calls utilized to solve each combinatorial problem instance. Table 4 gives the results of these experiments, where the number of function calls in CoT and Refine methods was explicitly controlled to make them equal to SGE function calls. The results analysis shows that the SGE method achieves a 27.84% better performance compared to the Decomposition method but requires 87.89% more function calls. Thus, while SGE offers superior performance, it does so at a marginally higher operational cost. Therefore, the application of this method is particularly justified in scenarios where performance gains are prioritized over cost efficiency.

### 5.3 Results on reasoning tasks

To evaluate the versatility of SGE in handling different types of tasks, we conducted experiments across eight datasets commonly referenced in LLM research, categorized into three distinct task types: arithmetic, commonsense reasoning, and symbolic reasoning. Table 3 gives the results of these experiments, with each dataset comprising train and test splits where SGE and baseline methods were applied to the test splits. The results analysis shows that the SGE method demonstrates incremental but consistently superior performance across all task categories. Notably, the method shows particular strength in arithmetic tasks, where it achieves an average improvement of 4.83%, compared to 1.24% in commonsense reasoning tasks and 1.32% in symbolic reasoning tasks. This demonstrates the method's applicability and effectiveness across a diverse range of tasks, extending beyond combinatorial problems.

## 6 Conclusion

This study has explored the application of Large Language Models to combinatorial problems, a category of tasks known for their NP-hardness. Our research introduced a 'Self-Guiding Exploration' prompting strategy that effectively utilizes the inherent strengths of LLMs. By generating multiple thought trajectories tailored to various CPs and autonomously decomposing them into manageable subtasks. Our findings confirm that SGE outperforms existing strategies, improving optimization performance by 27.84% and achieving a 2.46% higher accuracy in reasoning tasks. Notably, SGE shows a 34.85% smaller gap with the global optimum on complex tasks like the Job Sheduling Problem compared to baseline methods. These results underline the potential of advanced LLM strategies in complex problem-solving scenarios, suggesting that the right techniques can enhance the utility of LLMs in critical logistics and resource management applications.

Despite the performance improvements demonstrated by the SGE, several limitations have emerged that merit attention. Firstly, SGE performance depends on the choice of language model. Secondly, the operational costs associated with SGE are notably higher; it requires 87.89% more function calls than the Decomposition method. These issues present clear avenues for future research. Enhancing SGE's computational efficiency while maintaining its high performance could broaden its applicability and make it a more practical choice for a wider range of problems.

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

# A Appendix

## A.1 Baseline prompting techniques

Combinatorial problems are too complex for solving using direct approach. To solve in several shots, different methods can be used such as few-shot prompting, chain-of-thought, exploration-of-thought, decomposition, and self-refine advanced prompting techniques. Traditional few-shot prompting involves teaching an LLM to derive an answer $A$ to a query $Q$ using a limited set of contextual examples $D = \{E_1, ..., E_{|D|}\}$, where $A = f(Q, D)$. In the simplest few-shot setup, examples are formatted as $E_j = (Q_j, A_j)$, where each $Q_j, A_j$ are corresponding prompt and solution of example problem $j$. For Chain of Thought (CoT) prompting, the objective shifts to generating a sequence of intermediate reasoning steps, or "thoughts" $T$, and subsequently deriving the final answer from $T$. These in-context examples are structured as $E_j = (Q_j, (T_{j,1}, \ldots, T_{j,k}), A_j)$. Exploration-of-Thought techniques, such as those described in [37, 42], focus on dividing the problem $Q$ into $K$ subproblems and auto-generating thoughts $T_k$ through subproblem queries $Q_k$ with $T_k = f(Q_k, T_{k-1})$. The ultimate answer is then computed as $A = f(Q, T_K)$. In addition to that, in Tree-of-Thought (ToT) and Graph-of-Thought (GoT) prompting strategies, the problem is split into $N$ thought trajectories $T_k^n$, using pre-determined subqueries $Q_k^n$, with the final answer determined by $A = f(Q, T_K^1, ..., T_K^N)$. These strategies necessitate manually identifying effective subqueries $Q_k^n$ for each specific task (e.g., arithmetic, commonsense, symbolic). Decomposition prompting strategies, as referenced in [48, 12] get $Q_k^n$ subqueries through in-context examples formatted similarly to CoT exemplars: $E_j = \left( (Q_j, (Q_{j,1}, T_{j,1}), ..., (Q_{j,k_j}, T_{j,k_j})) A_j \right)$. Another prompting method called self-refinement [19] uses feedback and refine procedures to generate feedback queries $Q_{k_{feedback}}^n$ and subsequently refine thoughts $T_k^n$. In our research we combined these advanced techniques and updated them into new method called self-guiding exploration to enhance performance of LLMs for CPs.

## A.2 Combinatorial problems

Combinatorial problems are decision problems where solver needs to assign binary decision variable $x \in \{0, 1\}$ to minimize some cost function or maximize reward function given input $C$.

**Assignment problem.** Despite not being classified as NP-hard and solvable in polynomial time using the Hungarian algorithm, the Assignment Problem remains a fundamental combinatorial challenge. This problem entails optimally assigning $n$ tasks to $n$ workers, aiming to minimize the total cost or maximize the total efficiency of the assignments. The input array $C$ is represented as $n \times n$ cost matrix, where the element at the $i^{th}$ row and $j^{th}$ column represents the cost of assigning the $j^{th}$ task to the $i^{th}$ worker. The goal is essentially about finding a one-to-one matching between workers and tasks with the objective of minimizing the total cost, i.e.

$$\min\ g(x) = \sum_{i=1}^{n} \sum_{j=1}^{n} c_{ij} x_{ij}, \tag{1}$$

$$\text{s.t.} \sum_{i=1}^{n} x_{ij} = 1; \quad \sum_{j=1}^{n} x_{ij} = 1; \quad x_{ij} \in \{0, 1\} \quad \forall i, j. \tag{2}$$

**Knapsack problem.** The knapsack problem is a classic combinatorial problem, focusing on resource allocation. It is a decision problem in which the goal is to pack items from a set of items with given weights $w$ and values $v$ into a container with a maximum capacity $W$. The input array $C$ is represented as $n \times 2$ matrix with volume and value information of every item $i$. The goal is to maximize total value of packed items without exceeding container capacity, i.e.

$$\max\ g(x) = \sum_{i=1}^{n} v_i x_i, \tag{3}$$

$$\text{s.t.} \sum_{i=1}^{n} w_i x_i \leq W; \quad x_i \in \{0, 1\} \quad \forall i. \tag{4}$$

**Bin packing problem.** The bin packing problem is a combinatorial problem that involves efficiently packing $n$ objects of different sizes $w$ into a finite number of $k$ bin containers of fixed capacity $W$ in a way that minimizes the number of bins used. The input array $C$ is represented as $n \times 1$ vector with size information of every item $i$,

$$\min \; g(x) = \sum_{j=1}^{k} x_j, \tag{5}$$

$$\text{s.t.} \sum_{i=1}^{n} x_{ij} = 1; \quad \sum_{i=1}^{n} w_i x_{ij} \leq W; \quad x_{ij} \in \{0,1\} \quad \forall i, j. \tag{6}$$

**Travelling salesman problem.** In the travelling salesman problem, given a list of $n$ cities and the distances $d$ between each pair of cities, what is the shortest possible route that visits each city exactly once and returns to the origin city. The input array $C$ is represented as $n \times n$ cost matrix, where the element at the $i^{th}$ row and $j^{th}$ column represents the cost $d_{ij}$ of travelling between these two cities,

$$\min \; g(x) = \sum_{i=1}^{n} \sum_{j=1}^{n} d_{ij} x_{ij}, \tag{7}$$

$$\text{s.t.} \sum_{i=1}^{n} x_{ij} = 1; \quad \sum_{j=1}^{n} x_{ij} = 1; \quad x_{ij} \in \{0,1\} \quad \forall i, j. \tag{8}$$

**Vehicle routing problem.** The vehicle routing problem generalizes the TSP by incorporating multiple vehicles into the route planning. The VRP seeks to determine the optimal set of routes for a fleet of $K$ vehicles with maximum capacity $P$ to deliver goods to $n$ customers, typically from a central depot, with the objective of minimizing the total travel cost. In addition to TSP, the input $C$ also includes the demand of each customer, i.e.

$$\min \; g(x) = \sum_{i=1}^{n} \sum_{j=1}^{n} d_{ij} x_{ijk}, \tag{9}$$

$$\text{s.t.} \sum_{i=1}^{n} x_{ijk} = 1; \quad \sum_{j=1}^{n} x_{ijk} = 1; \quad x_{ijk} \in \{0,1\} \quad \forall i, j, k. \tag{10}$$

In cases when the capacity $P$ is less than the customer's demand, the vehicle will go between the depot and the customer until the demand is satisfied.

**Job scheduling problem.** The job scheduling problem focuses on scheduling $n$ jobs on $m$ machines, where each job $i$ consists of a sequence of $m$ operations that need to be processed in a specified order. Each operation requires a specific machine for a certain period of time, and each machine can handle only one operation at a time. The input array $C$ is represented as $n \times m \times 2$ cost matrix, which stores machine id and completion time for every operation $j$ of every job $i$. The primary objective is to minimize the makespan, which is the total time required to complete all jobs, effectively reducing the time from the start of the first operation to the completion of the last operation across all jobs.

**Challenges in solving combinatorial problems.** The difficulty of the combinatorial problems like TSP, VRP, JSP, and Knapsack problem stems from several intrinsic and mathematical characteristics common among them. These problems are typically classified as NP-hard, which fundamentally contributes to their computational complexity. As the number of elements (cities, jobs, items, etc.) increases, the number of possible combinations or permutations explodes exponentially. For instance, the TSP with $n$ cities has $(n-1)!$ possible routes to evaluate. This exponential growth means that the time required to examine all possible solutions becomes impractically long even for relatively small $n$. In addition to that combinatorial problems involve decisions that are interdependent, where the choice for one element affects the options and costs for others. For example, in the JSP, the order in which jobs are processed on one machine can affect the scheduling for other machines.

Table 5: Percentage Performance Improvement Compared to IO Prompting on Job Scheduling Problem. Columns Show the Number of $n$ Jobs and $m$ Machines.

|  | n50m10 | n50m20 | n100m10 | n100m20 |
|---|---|---|---|---|
| LNS | 57.2 | 59.1 | 59.6 | 60.8 |
| OR-Tools | 61.3 | 63.1 | 62.4 | 61.7 |
| SGE | 59.1 | 62.9 | 61.4 | 60.8 |

Table 6: Percentage Performance Improvement Compared to IO Prompting on Vehicle Routing Problem. Columns Show the Number of Nodes.

|  | n100 | n150 | n200 |
|---|---|---|---|
| LNS | 57.8 | 58.7 | 58.1 |
| OR-Tools | 62.5 | 61.2 | 60.3 |
| LKH3 | 65.3 | 64.4 | 65.8 |
| SGE | 59.6 | 60.1 | 59.8 |

**Applicability of SGE to combinatorial problems.**   For combinatorial problems, obtaining exact solutions is generally computationally prohibitive. Consequently, approximation algorithms and heuristic methods are frequently employed to tackle these challenges. Typically, these problems are broken down into more manageable subproblems. Initial solutions are then generated using heuristic functions, which are subsequently refined through additional heuristic techniques to enhance solution quality. The SGE approach is particularly well-suited for combinatorial problems as it embodies this multi-stage process: it systematically decomposes the problem, explores potential solving methods, and iteratively refines the solutions.

### A.3 Additional experiments and results

To further strengthen our results and provide a comprehensive comparison of our method with well-established solvers and heuristics, we conducted additional experiments on larger instances of combinatorial problems. Specifically, we included experiments with Job Shop Scheduling Problems (Table 5) and Vehicle Routing Problems (Table 6), comparing our approach against well-known solvers and heuristics. The goal was to evaluate the scalability of our method and assess its performance relative to commonly used combinatorial optimization methods.

Recognizing the need for understanding computational overhead, we have also provided an analysis of the computational cost involved in running our algorithm (Table 7). We have introduced a new baseline using large language models (LLMs) for heuristic generation, which is included in Table 8. We used the general version of EoH [17] (originally used for the Bin Packing problem). Specifically, in our case, for the Job Shop Scheduling task, EoH generated heuristics that scored the job nodes, and the algorithm then selected the job with the highest score as the next one in the schedule. However, we found that for more complex tasks than Bin Packing (e.g., TSP), EoH is better employed with Guided Local Search, where simple heuristics like swapping are used for local optimization, and EoH identifies a heuristic that can disturb the local optimum to explore a better region of the solution space. We believe that EoH built in this way, developing specific heuristic programs for each problem instance, would likely perform similarly to Google OR-Tools and achieve better performance. This approach effectively works as a metaheuristic enhancer, providing state-of-the-art results as seen in [17].

Table 7: VRP Average Total Cost.

| Number of Nodes | Total Cost |
|---|---|
| 5 | $0.0961 |
| 8 | $0.1676 |
| 12 | $0.1964 |
| 20 | $0.3515 |

Table 8: Percentage Performance Improvement Compared to IO Prompting on Job Scheduling Problem with New Baseline. Columns Show the Number of Nodes.

|       | n50m10 | n50m20 | n100m10 | n100m20 |
|-------|--------|--------|---------|---------|
| EoH   | 57.8   | 59.6   | 56.4    | 57.1    |
| SGE   | 59.1   | 62.9   | 61.4    | 60.8    |

Table 9: **Effect of problem size** on $SGE$ performance using `gpt-4` w/ code interpreter. The results are represented as performance percentage improvement compared to IO solution (the bigger it is the better).

| Size | Assignment | Knapsack | Bin Packing | TSP | VRP | JSP |
|------|------------|----------|-------------|-------|-------|-------|
| 5    | 44.05      | 75.94    | 90.92       | 76.77 | 80.47 | 87.81 |
| 8    | 42.03      | 77.14    | 75.92       | 80.12 | 80.78 | 71.68 |
| 12   | 42.65      | 72.50    | 80.36       | 79.93 | 76.18 | 77.08 |
| 15   | 41.78      | 71.70    | 71.61       | 73.24 | 78.41 | 79.73 |
| 20   | 40.24      | 67.21    | 69.78       | 67.14 | 65.19 | 76.29 |
| 25   | 39.83      | 67.07    | 66.35       | 60.17 | 60.87 | 69.52 |
| 30   | 38.75      | 61.15    | 68.13       | 67.32 | 61.55 | 65.20 |

Table 10: **Effect of model selection** on $SGE$ performance using `gpt-4` w/ code interpreter. The results are represented as performance percentage improvement compared to IO solution (the bigger it is the better).

| Task                 | GPT-3.5 | GPT-4 | Gemini-1.5 | Llama-2-7b | Llama-2-70b |
|----------------------|---------|-------|------------|------------|-------------|
| Assignment           | 25.35   | 41.33 | 40.46      | 23.19      | 24.01       |
| Knapsack             | 34.43   | 70.39 | 65.87      | 28.37      | 32.82       |
| Bin Packing          | 35.83   | 74.72 | 67.63      | 29.78      | 33.76       |
| Travelling Salesman  | 35.26   | 72.10 | 68.09      | 30.65      | 33.77       |
| Vehicle Routing      | 36.28   | 71.92 | 68.02      | 30.55      | 33.65       |
| Job Scheduling       | 35.02   | 75.33 | 67.89      | 30.39      | 34.42       |

Table 11: **Results for reasoning tasks** using `gpt-3.5` with a code interpreter are presented as accuracies on benchmark test sets.

| Method | Arithmetic | | | | Commonsense | | | Symbolic | Avg. |
|--------|------|------|-------|-------|------------|------|------|------------|------|
|        | AQUA | GSM8K | SVAMP | ASDiv | StrategyQA | CSQA | ARC | LastLetter | |
| IO Prompting | 44.16 | 57.05 | 60.02 | 65.43 | 54.70 | 55.68 | 64.35 | 56.21 | 57.20 |
| CoT Prompting | 58.35 | 75.84 | 80.30 | 87.39 | 72.97 | 74.36 | 85.48 | 74.63 | 76.17 |
| Refine Prompting | 58.81 | 76.10 | 80.44 | 87.35 | 73.38 | 74.13 | 85.71 | 74.78 | 76.34 |
| Decomp Prompting | 71.64 | 82.11 | 84.34 | 89.83 | 76.16 | 78.84 | 87.48 | 77.89 | 81.11 |
| Ours | **72.84** | **86.18** | **86.44** | **90.17** | **77.44** | **79.09** | **87.83** | **78.78** | **82.27** |

## A.4 Complete results of combinatorial problem experiments

Table 12: Comparison of various combinatorial problems based on their average cost per problem size, using `gpt-3.5`. The Knapsack problem aims to maximize returns, while other problems focus on minimizing costs.

| Size | Method | Assignment | Knapsack | Bin Packing | TSP | VRP | JSP |
|------|--------|-----------|----------|-------------|------|------|------|
| 5 NODES | Avg. | 102.23 | 710.53 | 13.480 | 679.26 | 809.67 | 3873.98 |
| | IO | 119.25 | 604.83 | 15.220 | 784.98 | 951.89 | 4499.38 |
| | CoT | 111.61 | 637.73 | 14.040 | 701.50 | 853.91 | 4032.66 |
| | Refine | 96.350 | 750.91 | 14.150 | 711.28 | 828.42 | 4048.06 |
| | Decomp | 94.350 | 738.32 | 12.920 | 643.25 | 746.36 | 3628.41 |
| | Ours | **89.610** | **820.85** | **11.080** | **555.29** | **667.76** | **3161.4** |
| 8 NODES | Avg. | 144.99 | 1053.98 | 23.080 | 1261.02 | 1436.94 | 8714.13 |
| | IO | 166.38 | 870.54 | 26.850 | 1425.48 | 1634.52 | 9896.3 |
| | CoT | 157.67 | 937.96 | 23.910 | 1341.56 | 1527.66 | 9202.0 |
| | Refine | 137.22 | 1118.63 | 23.500 | 1293.09 | 1504.8 | 9068.51 |
| | Decomp | 137.40 | 1122.79 | 21.670 | 1195.11 | 1327.99 | 8088.35 |
| | Ours | **126.27** | **1220.0** | **19.450** | **1049.85** | **1189.73** | **7315.47** |
| 12 NODES | Avg. | 218.82 | 1705.26 | 32.940 | 1695.44 | 2149.65 | 22142.9 |
| | IO | 257.33 | 1462.01 | 37.420 | 1914.84 | 2424.01 | 24954.7 |
| | CoT | 239.36 | 1553.15 | 34.380 | 1758.64 | 2305.24 | 23697.5 |
| | Refine | 204.69 | 1798.81 | 34.020 | 1775.3 | 2219.27 | 22721.6 |
| | Decomp | 205.31 | 1790.05 | 30.880 | 1589.09 | 1992.09 | 20553.1 |
| | Ours | **187.44** | **1922.26** | **27.970** | **1439.32** | **1807.66** | **18787.4** |
| 15 NODES | Avg. | 254.60 | 2412.26 | 37.230 | 2536.35 | 2704.09 | 38224.9 |
| | IO | 294.53 | 2010.03 | 42.810 | 2855.74 | 3070.15 | 42092.4 |
| | CoT | 281.95 | 2208.41 | 38.840 | 2743.79 | 2817.55 | 40660.3 |
| | Refine | 242.95 | 2537.45 | 38.710 | 2625.05 | 2807.14 | 40123.6 |
| | Decomp | 235.55 | 2606.89 | 34.770 | 2354.43 | 2584.13 | 36515.9 |
| | Ours | **218.01** | **2698.54** | **31.040** | **2102.76** | **2241.5** | **31732.2** |
| 20 NODES | Avg. | 323.14 | 2799.89 | 47.160 | 3122.83 | 3317.71 | 58569.7 |
| | IO | 377.67 | 2433.27 | 53.060 | 3490.25 | 3798.23 | 66151.8 |
| | CoT | 350.43 | 2523.88 | 50.330 | 3342.51 | 3504.49 | 62343.8 |
| | Refine | 311.98 | 2867.08 | 48.550 | 3202.12 | 3356.6 | 61383.3 |
| | Decomp | 298.35 | 2988.4 | 43.690 | 2924.58 | 3120.93 | 54426.8 |
| | Ours | **277.27** | **3186.8** | **40.180** | **2654.7** | **2808.3** | **48542.6** |
| 25 NODES | Avg. | 411.80 | 3732.26 | 14.640 | 3787.11 | 3991.53 | 103789.0 |
| | IO | 467.70 | 3208.8 | 16.740 | 4246.8 | 4470.97 | 117662.0 |
| | CoT | 449.08 | 3310.65 | 15.670 | 4023.17 | 4152.43 | 111329.0 |
| | Refine | 387.31 | 3854.93 | 15.150 | 3943.46 | 4101.81 | 105411.0 |
| | Decomp | 394.03 | 3988.58 | 13.560 | 3590.42 | 3832.48 | 97377.8 |
| | Ours | **360.87** | **4298.35** | **12.060** | **3131.7** | **3399.95** | **87164.7** |
| 30 NODES | Avg. | 512.38 | 4653.31 | 83.860 | 4995.58 | 5725.6 | 144036.0 |
| | IO | 595.35 | 3957.27 | 92.660 | 5605.85 | 6501.08 | 162140.0 |
| | CoT | 568.30 | 4162.98 | 88.780 | 5308.59 | 6051.78 | 147656.0 |
| | Refine | 481.16 | 4796.2 | 88.280 | 5108.91 | 5853.66 | 152181.0 |
| | Decomp | 476.20 | 5028.73 | 79.800 | 4762.89 | 5449.52 | 134230.0 |
| | Ours | **440.87** | **5321.36** | **69.790** | **4191.68** | **4771.98** | **123974.0** |

Table 13: Comparison of various combinatorial problems based on their average cost per problem size, using `gpt-4`. The Knapsack problem aims to maximize returns, while other problems focus on minimizing costs.

| Size | Method | Assignment | Knapsack | Bin Packing | TSP | VRP | JSP |
|------|--------|-----------|----------|-------------|-----|-----|-----|
| 5 NODES | Avg. | 198.69 | 370.00 | 7.4800 | 372.00 | 442.95 | 2131.42 |
| | IO | 267.83 | 267.03 | 9.5800 | 459.92 | 555.57 | 2701.22 |
| | CoT | 234.43 | 300.42 | 8.2000 | 415.99 | 490.05 | 2360.95 |
| | Refine | 174.66 | 400.32 | 8.1600 | 394.11 | 474.43 | 2259.75 |
| | Decomp | 166.67 | 412.43 | 6.4400 | 329.82 | 386.85 | 1896.89 |
| | Ours | **149.84** | **469.81** | **5.0200** | **260.18** | **307.84** | **1438.3** |
| 8 NODES | Avg. | 278.35 | 571.94 | 13.030 | 710.49 | 821.54 | 4878.37 |
| | IO | 365.34 | 415.19 | 16.160 | 890.27 | 1029.37 | 6024.32 |
| | CoT | 324.71 | 456.91 | 14.050 | 775.93 | 901.69 | 5359.79 |
| | Refine | 249.70 | 623.87 | 14.140 | 761.91 | 872.33 | 5163.49 |
| | Decomp | 240.23 | 628.28 | 11.600 | 630.08 | 734.92 | 4335.16 |
| | Ours | **211.78** | **735.45** | **9.1800** | **494.25** | **569.41** | **3509.08** |
| 12 NODES | Avg. | 405.56 | 944.25 | 18.730 | 962.85 | 1238.87 | 12501.5 |
| | IO | 534.81 | 680.40 | 23.640 | 1215.49 | 1537.63 | 15581.6 |
| | CoT | 475.38 | 791.10 | 20.760 | 1024.87 | 1337.63 | 13557.7 |
| | Refine | 359.52 | 1031.5 | 19.520 | 1034.36 | 1325.99 | 13395.1 |
| | Decomp | 351.35 | 1044.55 | 16.640 | 864.00 | 1120.36 | 11173.7 |
| | Ours | **306.72** | **1173.71** | **13.110** | **675.53** | **872.74** | **8799.38** |
| 15 NODES | Avg. | 478.16 | 1349.74 | 21.780 | 1469.81 | 1555.83 | 21885.7 |
| | IO | 634.70 | 996.41 | 27.150 | 1822.75 | 1958.65 | 27565.6 |
| | CoT | 561.76 | 1133.85 | 23.670 | 1608.72 | 1670.3 | 23247.9 |
| | Refine | 415.50 | 1460.7 | 22.820 | 1567.79 | 1639.87 | 23778.5 |
| | Decomp | 409.32 | 1446.91 | 19.420 | 1297.62 | 1412.5 | 19499.5 |
| | Ours | **369.55** | **1710.83** | **15.820** | **1052.15** | **1097.85** | **15336.9** |
| 20 NODES | Avg. | 587.33 | 1581.38 | 27.780 | 1822.89 | 1918.24 | 34243.4 |
| | IO | 754.98 | 1189.83 | 34.320 | 2262.86 | 2356.68 | 42589.9 |
| | CoT | 669.13 | 1326.23 | 30.270 | 1981.0 | 2065.07 | 36955.8 |
| | Refine | 536.41 | 1678.17 | 29.380 | 1902.42 | 1998.58 | 36585.5 |
| | Decomp | 525.00 | 1723.18 | 24.700 | 1614.36 | 1744.24 | 30926.3 |
| | Ours | **451.14** | **1989.48** | **20.220** | **1353.84** | **1426.64** | **24159.4** |
| 25 NODES | Avg. | 743.93 | 2148.32 | 8.6300 | 2249.6 | 2381.92 | 61530.6 |
| | IO | 959.89 | 1622.78 | 10.430 | 2690.03 | 2872.54 | 75770.1 |
| | CoT | 843.97 | 1827.16 | 9.3000 | 2430.33 | 2563.55 | 67624.2 |
| | Refine | 678.44 | 2238.86 | 9.3300 | 2445.45 | 2510.81 | 64490.3 |
| | Decomp | 659.79 | 2341.61 | 7.8200 | 2002.71 | 2177.05 | 55070.7 |
| | Ours | **577.56** | **2711.18** | **6.2700** | **1679.49** | **1785.68** | **44697.5** |
| 30 NODES | Avg. | 912.48 | 2707.35 | 50.790 | 3070.36 | 3388.03 | 87633.7 |
| | IO | 1166.32 | 2079.35 | 62.520 | 3735.78 | 4104.37 | 104798.0 |
| | CoT | 1049.36 | 2254.12 | 55.550 | 3296.34 | 3602.09 | 95887.8 |
| | Refine | 844.18 | 2932.01 | 52.290 | 3278.37 | 3546.61 | 93372.0 |
| | Decomp | 788.12 | 2920.33 | 46.390 | 2808.58 | 3146.44 | 80674.5 |
| | Ours | **714.42** | **3350.95** | **37.180** | **2232.74** | **2540.65** | **63435.8** |

Table 14: Comparison of various combinatorial problems based on their average cost per problem size, using `gemini-1.5`. The Knapsack problem aims to maximize returns, while other problems focus on minimizing costs.

| Size | Method | Assignment | Knapsack | Bin Packing | TSP | VRP | JSP |
|------|--------|-----------|----------|-------------|-----|-----|-----|
| 5 NODES | Avg. | 185.01 | 395.98 | 7.8900 | 392.96 | 475.25 | 2247.28 |
| | IO | 249.50 | 293.42 | 9.7400 | 488.09 | 581.55 | 2781.78 |
| | CoT | 212.33 | 325.58 | 8.6400 | 432.11 | 530.63 | 2460.15 |
| | Refine | 161.90 | 424.53 | 8.4800 | 414.98 | 500.49 | 2410.6 |
| | Decomp | 161.13 | 433.07 | 7.0600 | 346.36 | 432.60 | 2007.1 |
| | Ours | **140.19** | **503.30** | **5.5400** | **283.29** | **330.98** | **1576.76** |
| 8 NODES | Avg. | 262.29 | 596.51 | 13.650 | 739.72 | 853.62 | 5263.2 |
| | IO | 348.38 | 450.87 | 17.040 | 917.04 | 1055.48 | 6532.72 |
| | CoT | 296.24 | 489.46 | 14.860 | 803.14 | 938.79 | 5783.3 |
| | Refine | 238.56 | 637.94 | 14.520 | 781.85 | 903.12 | 5646.28 |
| | Decomp | 230.29 | 653.73 | 12.050 | 672.67 | 769.21 | 4636.82 |
| | Ours | **197.98** | **750.57** | **9.7700** | **523.89** | **601.50** | **3716.89** |
| 12 NODES | Avg. | 382.15 | 1005.24 | 19.690 | 1012.68 | 1291.47 | 13134.0 |
| | IO | 490.19 | 729.67 | 24.330 | 1249.43 | 1583.87 | 15971.6 |
| | CoT | 437.93 | 848.80 | 21.730 | 1080.36 | 1396.44 | 14301.9 |
| | Refine | 346.04 | 1068.12 | 20.700 | 1077.78 | 1360.73 | 14185.8 |
| | Decomp | 339.04 | 1129.44 | 17.490 | 927.75 | 1194.11 | 11726.4 |
| | Ours | **297.53** | **1250.18** | **14.200** | **728.06** | **922.21** | **9484.31** |
| 15 NODES | Avg. | 444.10 | 1449.26 | 23.150 | 1564.27 | 1653.6 | 23061.3 |
| | IO | 585.87 | 1090.76 | 27.950 | 1940.9 | 2036.78 | 28629.8 |
| | CoT | 509.47 | 1216.37 | 25.330 | 1658.02 | 1801.66 | 25246.4 |
| | Refine | 399.71 | 1541.78 | 24.940 | 1687.24 | 1715.89 | 23949.6 |
| | Decomp | 385.44 | 1585.02 | 20.670 | 1416.65 | 1510.22 | 20343.8 |
| | Ours | **340.02** | **1812.35** | **16.830** | **1118.55** | **1203.46** | **17137.1** |
| 20 NODES | Avg. | 548.21 | 1678.75 | 29.180 | 1945.87 | 2009.98 | 35753.1 |
| | IO | 701.27 | 1280.46 | 35.360 | 2396.6 | 2438.29 | 44349.1 |
| | CoT | 620.99 | 1428.82 | 31.960 | 2111.46 | 2166.27 | 38059.4 |
| | Refine | 497.49 | 1810.0 | 30.120 | 2025.49 | 2142.69 | 37385.7 |
| | Decomp | 480.97 | 1800.23 | 26.790 | 1740.08 | 1806.83 | 32493.8 |
| | Ours | **440.31** | **2074.24** | **21.690** | **1455.74** | **1495.83** | **26477.4** |
| 25 NODES | Avg. | 708.54 | 2233.73 | 9.2200 | 2352.5 | 2496.48 | 64955.1 |
| | IO | 911.14 | 1694.31 | 11.280 | 2890.44 | 2978.91 | 77142.4 |
| | CoT | 791.74 | 1878.46 | 9.9700 | 2504.54 | 2720.89 | 70181.8 |
| | Refine | 642.79 | 2401.42 | 9.7100 | 2478.42 | 2670.85 | 69104.9 |
| | Decomp | 634.67 | 2434.54 | 8.4200 | 2126.79 | 2266.56 | 60013.5 |
| | Ours | **562.34** | **2759.92** | **6.7300** | **1762.3** | **1845.18** | **48333.0** |
| 30 NODES | Avg. | 863.39 | 2855.59 | 52.430 | 3200.01 | 3534.72 | 91402.1 |
| | IO | 1108.6 | 2189.92 | 62.270 | 3807.69 | 4232.51 | 111716.0 |
| | CoT | 1002.04 | 2442.99 | 56.720 | 3411.03 | 3757.36 | 96055.7 |
| | Refine | 784.26 | 2991.77 | 55.310 | 3447.36 | 3764.56 | 95672.6 |
| | Decomp | 751.20 | 3134.08 | 47.690 | 2886.03 | 3265.4 | 83939.4 |
| | Ours | **670.85** | **3519.17** | **40.140** | **2447.95** | **2653.76** | **69626.6** |

Table 15: Comparison of various combinatorial problems based on their average cost per problem size, using `llama-2-7b`. The Knapsack problem aims to maximize returns, while other problems focus on minimizing costs.

| Size | Method | Assignment | Knapsack | Bin Packing | TSP | VRP | JSP |
|------|--------|-----------|----------|-------------|-----|-----|-----|
| 5 NODES | Avg. | 89.950 | 802.42 | 15.550 | 775.06 | 926.24 | 4397.27 |
| | IO | 104.24 | 698.34 | 17.110 | 853.58 | 1041.09 | 4958.95 |
| | CoT | 98.820 | 734.35 | 16.520 | 821.09 | 979.18 | 4575.45 |
| | Refine | 84.600 | 845.41 | 15.890 | 787.03 | 939.69 | 4634.51 |
| | Decomp | 82.460 | 839.18 | 14.810 | 745.98 | 873.03 | 4156.86 |
| | **Ours** | **79.630** | **894.84** | **13.410** | **667.62** | **798.20** | **3660.59** |
| 8 NODES | Avg. | 126.29 | 1197.13 | 26.320 | 1418.23 | 1645.42 | 9991.67 |
| | IO | 147.10 | 1013.08 | 28.850 | 1566.87 | 1832.76 | 11162.0 |
| | CoT | 136.66 | 1092.28 | 27.860 | 1498.77 | 1709.73 | 10389.7 |
| | Refine | 119.83 | 1252.94 | 27.120 | 1499.56 | 1710.88 | 10509.7 |
| | Decomp | 118.10 | 1274.02 | 25.100 | 1343.11 | 1572.56 | 9305.07 |
| | **Ours** | **109.75** | **1353.31** | **22.700** | **1182.83** | **1401.16** | **8591.86** |
| 12 NODES | Avg. | 191.22 | 1966.31 | 37.090 | 1921.87 | 2465.52 | 24978.0 |
| | IO | 220.40 | 1731.58 | 40.930 | 2146.2 | 2697.85 | 27795.8 |
| | CoT | 204.53 | 1810.85 | 39.550 | 1984.84 | 2596.84 | 25572.4 |
| | Refine | 181.54 | 2045.48 | 38.930 | 1992.89 | 2594.67 | 26383.4 |
| | Decomp | 178.63 | 2063.66 | 34.910 | 1834.9 | 2338.43 | 24007.4 |
| | **Ours** | **170.98** | **2179.96** | **31.130** | **1650.52** | **2099.78** | **21131.3** |
| 15 NODES | Avg. | 228.48 | 2739.37 | 42.610 | 2889.57 | 3017.02 | 43004.7 |
| | IO | 256.26 | 2351.63 | 48.170 | 3212.82 | 3331.89 | 47876.6 |
| | CoT | 250.10 | 2553.18 | 44.260 | 3025.74 | 3113.82 | 45938.3 |
| | Refine | 220.66 | 2843.52 | 44.470 | 3039.47 | 3077.27 | 43884.0 |
| | Decomp | 216.12 | 2896.08 | 40.350 | 2695.41 | 2901.24 | 41025.1 |
| | **Ours** | **199.24** | **3052.45** | **35.790** | **2474.42** | **2660.9** | **36299.3** |
| 20 NODES | Avg. | 287.07 | 3209.13 | 52.710 | 3548.1 | 3703.92 | 65481.9 |
| | IO | 329.28 | 2836.21 | 58.190 | 3983.11 | 4240.29 | 72279.5 |
| | CoT | 307.87 | 2933.81 | 54.780 | 3746.41 | 3837.66 | 69172.2 |
| | Refine | 277.27 | 3249.57 | 55.810 | 3667.07 | 3743.99 | 69032.1 |
| | Decomp | 268.09 | 3413.37 | 50.130 | 3373.55 | 3477.77 | 61580.1 |
| | **Ours** | **252.84** | **3612.7** | **44.640** | **2970.35** | **3219.91** | **55345.8** |
| 25 NODES | Avg. | 368.58 | 4255.19 | 16.310 | 4292.01 | 4503.47 | 116702.0 |
| | IO | 425.08 | 3693.28 | 18.060 | 4674.8 | 5083.89 | 127229.0 |
| | CoT | 395.84 | 3938.03 | 16.780 | 4529.96 | 4718.81 | 121452.0 |
| | Refine | 356.35 | 4438.0 | 17.280 | 4418.16 | 4667.3 | 123046.0 |
| | Decomp | 344.61 | 4534.51 | 15.410 | 4140.85 | 4267.0 | 110197.0 |
| | **Ours** | **321.02** | **4672.15** | **14.030** | **3696.27** | **3780.34** | **101586.0** |
| 30 NODES | Avg. | 452.94 | 5279.99 | 93.580 | 5711.01 | 6309.75 | 160913.0 |
| | IO | 504.64 | 4594.52 | 103.15 | 6409.08 | 7164.38 | 177461.0 |
| | CoT | 502.24 | 4891.09 | 98.630 | 5995.11 | 6626.95 | 166083.0 |
| | Refine | 431.79 | 5446.81 | 96.390 | 5957.74 | 6419.75 | 169324.0 |
| | Decomp | 426.99 | 5619.47 | 89.550 | 5402.13 | 5940.75 | 153129.0 |
| | **Ours** | **399.05** | **5848.08** | **80.200** | **4791.01** | **5396.94** | **138570.0** |

Table 16: Comparison of various combinatorial problems based on their average cost per problem size, using `llama-2-70b`. The Knapsack problem aims to maximize returns, while other problems focus on minimizing costs.

| Size | Method | Assignment | Knapsack | Bin Packing | TSP | VRP | JSP |
|------|--------|-----------|----------|-------------|-----|-----|-----|
| 5 NODES | Avg. | 96.760 | 757.12 | 14.650 | 719.93 | 864.61 | 4130.53 |
| | IO | 111.85 | 632.09 | 16.800 | 831.07 | 980.70 | 4706.85 |
| | CoT | 105.51 | 700.25 | 15.590 | 744.12 | 917.25 | 4290.67 |
| | Refine | 92.620 | 788.01 | 15.130 | 740.45 | 897.79 | 4297.34 |
| | Decomp | 89.850 | 815.64 | 13.340 | 689.33 | 807.57 | 3915.78 |
| | Ours | **83.980** | **849.63** | **12.410** | **594.70** | **719.74** | **3442.0** |
| 8 NODES | Avg. | 134.41 | 1140.45 | 24.710 | 1321.35 | 1538.71 | 9429.18 |
| | IO | 155.54 | 967.89 | 28.290 | 1477.89 | 1768.47 | 10645.8 |
| | CoT | 145.07 | 1035.49 | 25.540 | 1399.97 | 1602.0 | 9988.41 |
| | Refine | 128.91 | 1186.02 | 26.100 | 1355.37 | 1601.33 | 9835.02 |
| | Decomp | 126.39 | 1209.13 | 23.090 | 1263.55 | 1424.77 | 8794.53 |
| | Ours | **116.14** | **1303.73** | **20.560** | **1109.98** | **1296.99** | **7882.13** |
| 12 NODES | Avg. | 201.99 | 1834.2 | 35.110 | 1805.11 | 2315.91 | 23294.2 |
| | IO | 231.77 | 1562.82 | 40.340 | 2081.41 | 2610.96 | 26535.8 |
| | CoT | 220.85 | 1690.42 | 36.560 | 1922.4 | 2462.96 | 24542.6 |
| | Refine | 190.52 | 1865.06 | 36.340 | 1831.0 | 2363.75 | 23820.7 |
| | Decomp | 192.52 | 1937.35 | 32.570 | 1666.65 | 2169.12 | 21807.0 |
| | Ours | **174.27** | **2115.36** | **29.750** | **1524.11** | **1972.79** | **19764.7** |
| 15 NODES | Avg. | 239.40 | 2575.28 | 40.140 | 2728.94 | 2902.12 | 40452.0 |
| | IO | 271.97 | 2237.81 | 45.310 | 3091.61 | 3270.84 | 46491.8 |
| | CoT | 255.93 | 2296.4 | 41.940 | 2852.78 | 3034.48 | 42731.8 |
| | Refine | 234.52 | 2625.13 | 41.180 | 2850.11 | 2987.35 | 42280.3 |
| | Decomp | 226.70 | 2761.56 | 37.990 | 2533.08 | 2736.93 | 37362.8 |
| | Ours | **207.87** | **2955.51** | **34.280** | **2317.12** | **2480.98** | **33393.3** |
| 20 NODES | Avg. | 301.13 | 2972.25 | 50.240 | 3334.42 | 3499.69 | 61957.2 |
| | IO | 341.58 | 2565.48 | 56.260 | 3758.49 | 4025.37 | 69824.9 |
| | CoT | 330.69 | 2705.18 | 51.940 | 3462.22 | 3620.96 | 63469.1 |
| | Refine | 282.33 | 3031.65 | 52.210 | 3438.04 | 3567.26 | 64022.3 |
| | Decomp | 287.94 | 3116.33 | 47.640 | 3167.42 | 3297.34 | 58800.7 |
| | Ours | **263.10** | **3442.61** | **43.130** | **2845.92** | **2987.54** | **53669.1** |
| 25 NODES | Avg. | 389.69 | 4024.08 | 15.510 | 4029.51 | 4184.99 | 110414.0 |
| | IO | 443.99 | 3518.82 | 17.390 | 4600.6 | 4735.73 | 126686.0 |
| | CoT | 418.18 | 3677.69 | 15.880 | 4148.72 | 4358.13 | 115617.0 |
| | Refine | 373.97 | 4197.63 | 16.380 | 4136.03 | 4306.45 | 111541.0 |
| | Decomp | 369.43 | 4170.69 | 14.700 | 3822.3 | 3975.41 | 104862.0 |
| | Ours | **342.89** | **4555.58** | **13.180** | **3439.89** | **3549.23** | **93361.4** |
| 30 NODES | Avg. | 490.92 | 5026.68 | 88.930 | 5310.47 | 5986.86 | 154018.0 |
| | IO | 562.14 | 4332.75 | 99.580 | 5902.09 | 6848.47 | 171767.0 |
| | CoT | 535.73 | 4629.8 | 92.530 | 5670.09 | 6113.72 | 158276.0 |
| | Refine | 472.51 | 5217.75 | 92.730 | 5409.26 | 6165.51 | 160831.0 |
| | Decomp | 455.30 | 5339.41 | 85.010 | 4948.58 | 5562.01 | 146910.0 |
| | Ours | **428.95** | **5613.7** | **74.780** | **4622.34** | **5244.57** | **132308.0** |

## Example of VRP Prompting

- **Task Description**: You are given a list of customers with coordinates:
  - (1): (87, 39);
  - (2): (1, 90);
  - (3): (28, 65);
  - (4): (85, 85);

  and a list of customer demands:
  - (1): 12;
  - (2): 8;
  - (3): 16;
  - (4): 5.

  There is a depot with coordinates (50, 50) and one vehicle with a maximum capacity of 50. The goal is to find the route that has the minimum total length and go through all the customers, starting and ending at the depot.

## Example of VRP Exploration

- **Prompt**: [Task Description]. List heuristic methods to solve this problem. Return only method names separated by new lines.
- **Exploration Output**:
  - Nearest Neighbor Algorithm
  - Savings Algorithm
  - Sweep Algorithm
  - Clarke and Wright Algorithm
  - Genetic Algorithm
  - Simulated Annealing
  - Tabu Search
  - Ant Colony Optimization
  - Particle Swarm Optimization

## Example of VRP Decomposition

- **Prompt**: [Task Description]. Nearest Neighbor Algorithm. List the steps to use this heuristic method. Return only the steps, separated by new lines.
- **Decomposition Output**:
  - **Initialize at the Depot:** The starting point is the depot located at coordinates (50, 50).
  - **Find the Nearest Customer:** Calculate the Euclidean distance from the depot to each customer, and move to the closest one.
  - **Mark the Customer as Visited:** Once a customer is visited, they should not be revisited.
  - **Repeat Finding and Moving:** Continue finding the nearest unvisited customer from the current location until all customers have been visited.
  - **Return to the Depot:** After visiting the last customer, calculate the distance back to the depot to complete the route.
  - **Calculate the Total Length of the Route:** Sum up all the distances traveled from start to finish.

## Example of Subtask Resolution, Step One

- **Prompt**: [Task Description]. [Apply the heuristic's step one].
- **Subtask Resolution Output**:

  Initialize at the Depot
  - Current location: Depot at (50, 50).

## Example of Subtask Resolution, Step Two

- **Prompt**: [Task Description]. [Apply the heuristic's step two].
- **Subtask Resolution Output**:

  Find the Nearest Customer
  – Distance to Customer 1 at (87, 39): $\sqrt{(87-50)^2 + (39-50)^2}$
  – Distance to Customer 2 at (1, 90): $\sqrt{(1-50)^2 + (90-50)^2}$
  – Distance to Customer 3 at (28, 65): $\sqrt{(28-50)^2 + (65-50)^2}$
  – Distance to Customer 4 at (85, 85): $\sqrt{(85-50)^2 + (85-50)^2}$

  The initial distances from the depot to each customer are as follows:
  – Customer 1: 38.60
  – Customer 2: 63.25
  – Customer 3: 26.63 (nearest)
  – Customer 4: 49.50

## Example of Subtask Resolution, Step Three

- **Prompt**: [Task Description]. [Apply the heuristic's step three].
- **Subtask Resolution Output**:

  Mark the Customer as Visited.
  The nearest customer is Customer 3 at coordinates (28, 65). We will visit Customer 3 first.

## Example of Subtask Resolution, Step Six

- **Prompt**: [Task Description]. [Apply the heuristic's step six].
- **Subtask Resolution Output**:

  Calculate the Total Length of the Route.
  The complete route, following the Nearest Neighbor Algorithm from the depot, visiting all customers, and returning to the depot, totals a distance of approximately 232.22 units.
  Here's a summary of the route taken:

  – Start at Depot (50, 50)
  – Visit Customer 3 (28, 65)
  – Visit Customer 2 (1, 90)
  – Visit Customer 4 (85, 85)
  – Visit Customer 1 (87, 39)
  – Return to Depot (50, 50)

### Example of Feedback Prompting

- **Prompt**: [Task Description]. [Nearest Neighbor candidate solution]. Give feedback to the proposed solution.
- **Feedback Output**:

  Implement route optimization post-processing, such as 2-opt or 3-opt techniques, which iteratively refine the route by removing crossings and reducing travel distance.

### Example of Refinement Prompting

- **Prompt**: [Task Description]. [Nearest Neighbor candidate solution]. [Feedback]. Improve the solution given feedback.
- **Refinement Output**:
  - we'll apply a 2-opt optimization to the route. This will involve swapping segments of the route to find a shorter path. Given the route '1 -> 2 -> 3 -> 4', we'll try different swaps and see if we can reduce the total distance further.
  - Let's simulate the 2-opt optimization on this route.
  - After applying the 2-opt optimization technique, the final optimized route has been adjusted to: Customer 1 -> Customer 3 -> Customer 2 -> Customer 4, then returning to the depot. This rearrangement was made to minimize the total distance traveled.

### Example of Solution Integration

- **Prompt**: [Task Description]. [All candidate solutions]. Integrate all previous findings and provide the final answer. Return the answer only as a Python list of elements.
- **Refinement Output**: [1, 3, 2, 4]

