# OpenReview forum: "Self-Guiding Exploration for Combinatorial Problems"
_NeurIPS.cc/2024/Conference — NeurIPS 2024 poster_

### Official Review · Reviewer_8LKr · 2024-06-14

**Soundness:** 3
**Presentation:** 3
**Contribution:** 2
**Rating:** 6
**Confidence:** 3

**Summary:**

This paper introduces a self-guiding exploration (SGE) framework that integrates exploration-of-thought, decomposition, and refinement prompting strategies to tackle NP-hard combinatorial optimization problems (COPs). The experiments, conducted on various COPs and several reasoning tasks, demonstrate improved performance compared to existing prompt strategies.

*Based on my expertise, I will evaluate this paper from the perspective of combinatorial optimization.*

**Strengths:**

* The proposed SGE is general and innovative.
* The paper is well-organized and clearly written.
* The scope of empirical evaluation is extensive, including combinatorial optimization problems and various reasoning tasks.
* The source code is provided.

**Weaknesses:**

* Figure 2 is lacking in detail. A comprehensive example, including detailed prompts and outputs, should be provided.
* It appears that the decomposition process cannot be parallelized across subtasks due to the dependence on preceding $T_{k-1}^n$.
* Limited review of related work:
  * LLMs/FMs in CP: Many recent studies are missing. Please see [5] for a comprehensive review.
  * Learning-based methods in CP: Numerous NCO studies, especially those published in the last three years, are missing as well, please see [6].
* My main concern about this paper is the weak experiments, which significantly weaken the contribution of this paper.
  * This paper only considers 5-30 nodes, which is **too small** for the NCO community. I would suggest generating code, like [2-4], in the decomposition or subtask resolution stages if the performance on large-scale instances is inferior.
  * The optimality gap (w.r.t. the traditional solver, e.g., LKH3, OR-Tools) and inference time should be reported.
  * Missing baselines, such as [1-4]. Only several prompt strategies are chosen as baselines.

```tex
[1] Large Language Models as Evolutionary Optimizers
[2] Mathematical discoveries from program search with large language models
[3] Evolution of Heuristics: Towards Efficient Automatic Algorithm Design Using Large Language Model
[4] Large Language Models as Hyper-Heuristics for Combinatorial Optimization
[5] https://github.com/ai4co/awesome-fm4co
[6] https://github.com/Thinklab-SJTU/awesome-ml4co
```

**Questions:**

N/A

**Limitations:**

Yes, the authors discuss the limitations, such as more function calls, and the sensitivity to the used LLMs.

---

> ### Author Rebuttal · Authors · 2024-08-05
>
> __Comment:__ Figure 2 is lacking in detail. A comprehensive example, including detailed prompts and outputs, should be provided.
>
> __Response:__ Thank you for your feedback. Regarding this comment, we aimed to provide a comprehensive textual example, including prompts and model outputs, exactly in Figure 3. To clarify this concern, the first box of Figure 3 shows the text of the prompt that includes the TSP problem definition and the exact output from the Exploration stage of our method. The [Return Condition] represents the prompt text specific to the exploration stage, as discussed in Section 4.1 of the paper. These [Return Conditions] are general prompts that can be applied to various problems without modifications.
>
> The third box again repeats the problem definition, this time joined with the candidate solution. If this figure remains unclear, we added a new figure with detailed examples of prompts and outputs solving a VRP instance, both in the paper and in the rebuttal PDF. We hope this explanation clarifies the intent behind Figure 3 and addresses your concerns.
>
>
> __Comment:__ It appears that the decomposition process cannot be parallelized across subtasks due to the dependence on preceding Tk−1n.
>
> __Response:__ Thank you for your observation. Regarding this comment, the decomposition process is inherently sequential by design. The tasks within a single decomposition are meant to be executed in sequence, not in parallel. For example, Task $T_{k-1}^n$ might involve computing the distance from the current node to all other nodes, followed by Task $T_{k}^n$, which chooses the closest node as the next node to go to.
>
> These steps are dependent on each other and thus cannot be parallelized. However, processes from different decompositions can be parallelized. For instance, if one trajectory solves the VRP using the Ant Colony Optimization method and another uses the Nearest Neighbor approach, these processes are independent and can be executed in parallel.
>
> __Comment:__ Limited review of related work:
>
> __Response:__ Thank you for pointing this out. Regarding this comment, we initially did not include certain papers in our review because we believed they focused on different approaches, specifically tuning heuristics to specific instances of combinatorial problems. We now recognize the importance of including these related works to provide a more comprehensive literature review. We have extended our review to include these papers [1-6], acknowledging their relevance and contributions to the field. We hope this expanded literature review addresses your concern and provides a more complete context for our work. Thank you again.
>
>
> __Comment:__ This paper only considers 5-30 nodes, which is too small for the NCO community. I would suggest generating code, like [2-4], in the decomposition or subtask resolution stages if the performance on large-scale instances is inferior.
>
> __Response:__ Thank you for your comment. We initially focused on small problem sizes due to lower computational costs and the ability to obtain optimal solutions. The models without code interpreter are indeed inferior on large scale, but GPT4 with code interpreter can still show  good results. Based on your feedback, we have extended our experiments to include larger instances. To evaluate the performance of SGE against other state-of-the-art methods, we conducted experiments on larger problem sizes across various combinatorial problem domains. Table 1 (rebuttal) presents results on Job Shop Scheduling problems with 50 and 100 jobs, while Table 2 (rebuttal) shows experiments on the Vehicle Routing Problem with 100, 150, and 200 nodes. The results analysis shows that SGE performs better than LNS but falls short compared to LKH3 and Google OR-Tools, which are specifically tailored for combinatorial tasks. However, SGE remains applicable to a broader range of tasks (e.g. reasoning tasks). We believe that SGE's performance could improve further once libraries like LKH3 and Google OR-Tools are integrated into GPT-4's code interpreter, allowing our algorithm to leverage these tools within its solution trajectories.
>
>
> __Comment:__ The optimality gap (w.r.t. the traditional solver, e.g., LKH3, OR-Tools) and inference time should be reported.
>
> __Response:__ We appreciate your suggestion. In our original submission, we provided a comparison between SGE and Google OR-Tools, as shown in Table 2, focusing on small-size problems where globally optimal solutions could be achieved through exhaustive depth-first search. However, to further address your comment and enhance our analysis, we have conducted additional experiments, as mentioned earlier. The results from these larger-scale experiments (Tables 1 and 2), now included in our paper.
>
>
> __Comment:__ Missing baselines, such as [1-4]. Only several prompt strategies are chosen as baselines.
>
> __Response:__ Thank you for pointing this out. Regarding this comment, we acknowledge the importance of including relevant baselines. We assume that [2,3] may perform better than our approach on the Bin Packing problem, but according to [2], this is possible when a diverse list of programs (heuristics) is available as a database to avoid local minima. However, our goal was to develop an algorithm that does not rely on pre-existing examples of good solutions.
>
> To clarify further, we believe that using [3] (EoH) as a baseline is sufficient, as EoH is an improved version of [1,2]. We chose not to use [4] as a baseline, because it was published very recently, just before the NeurIPS deadline (although it's a good model). However, we have extended our paper to include the EoH baseline and have run its implementation on Job Shop Scheduling (JSP) problems. Table 4 (rebuttal) shows the results of SGE versus EoH. The analysis indicates that while the results are close, SGE achieved slightly better performance. We hope this additional comparison addresses your concern.

---

> > ### Comment · Reviewer_8LKr · 2024-08-08
> >
> > Thanks for your responses. It is surprising that SGE can outperform EOH (Table 4), which is the current SOTA based on code generation. Could the author elaborate on why this might be the case?

---

> ### Author Response · Authors · 2024-08-08
>
> Thank you for your question. We used general version of EoH (that was used for Bin Packing problem). Specifically, in our case, for the Job Shop Scheduling task, EoH generated heuristics that scored the job nodes, and the algorithm then selected the job with the highest score as the next one in the schedule. However, we found out, for more complex tasks than Bin Packing (e.g. TSP), EoH is better employed with the Guided Local Search, where simple heuristics like swapping are used for local optimization, and EoH identifies a heuristic that can disturb the local optimum to explore a better region of the solution space. We believe that EoH that is built this way and is developing specific heuristic programs for each problem instance would likely perform similarly to Google OR-Tools and achieve better performance. This way it works more like a metaheuristic enhancer and will give SOTA results like in [4]. Thank you again for your question, we will add this duscussion to our paper as well.

---

> > ### Comment · Reviewer_8LKr · 2024-08-09
> >
> > Thanks for your response, which addresses my concerns. I decided to raise my score.

---

> > > ### Author Response · Authors · 2024-08-09
> > >
> > > Thank you for your valuable feedback. We greatly appreciate your support and are pleased that our revisions have addressed your concerns.

---

### Official Review · Reviewer_auHA · 2024-07-11

**Soundness:** 3
**Presentation:** 3
**Contribution:** 3
**Rating:** 6
**Confidence:** 2

**Summary:**

The paper proposes an LLM-based solution for solving standard combinatorial search problems such as TSP and VRP. The proposed solution, called SGE, uses LLMs to (1) propose alternative approaches to solve the problem at hand, (2) decompose a chosen solution approach into subtasks, (3) identify if a given subtask is easy or hard, (4) if it’s easy, just do it, else (5) recursively call SGE to solve the given subtask. Finally, SGE also uses an LLM to integrate the results returned by the above queries.
Experimental results compared to other LLM-based approaches show huge gains for SGE.

**Strengths:**

- Solving combinatorial search problems is a core problem in AI
- SGE performs well experimentally
- The paper is, in general, well written
- The concept of having the LLM identify if a substask is hard or not, is neat

**Weaknesses:**

1.	It is not very clear to me why would one want to use an LLM to solve combinatorial problems.
2.	It seems that SGE is similar to prior work noted by the reviewers for solving these problems via intelligent prompting. The main difference I could see is the part where it checks if a task is easy or not and calls recursively afterwards, and the integration of all the results together.

The authors provide reasonable responses to both concerns.

**Questions:**

1.	Did you consider comparing SGE with a fast suboptimal algorithm for solving the evaluated CPs, e.g., LNS?
2.	Why is the Integrate step done by the LLMs and not directly by parsing and analyzing the answer so far?

**Limitations:**

Not releavnt.

---

> ### Author Rebuttal · Authors · 2024-08-05
>
> __Comment:__ It is not very clear to me why would one want to use an LLM to solve combinatorial problems.
>
> __Response:__ Thank you for your comment. The use of large language models to solve combinatorial problems offers an advantage by choosing a specific set of heuristics/metaheuristics tailored to concrete situations. Essentially, LLMs can function as highly flexible metaheuristics that possess knowledge of various heuristics and metaheuristics. As LLMs continue to advance, we expect that their ability to produce effective solutions in this domain wil improve.
>
>
> __Comment:__ It seems that SGE is similar to prior work noted by the reviewers for solving these problems via intelligent prompting. The main difference I could see is the part where it checks if a task is easy or not and calls recursively afterwards, and the integration of all the results together.
>
> __Response:__ Thank you for your comment. Regarding this observation, we acknowledge that different papers have explored various high-level prompting strategies. In our paper, we combined three of these approaches into a unified framework. However, to clarify your comment further, our method differs from prior work in that it is fully self-guiding. Unlike previous methods that rely on in-context learning with specific examples or prompts tailored to each problem, our approach does not require such modifications from problem to problem. (Please refer to the differences in the answer $A$ equations in Section 4.1 and A.1 (our approach does not require exemplars $E$)).
>
> The primary aim was to evaluate whether this self-guiding strategy could effectively improve performance on complex tasks while also maintaining strong results in reasoning tasks. We hope this addresses your concern and provides a clearer understanding of our approach.
>
>
> __Comment:__ Did you consider comparing SGE with a fast suboptimal algorithm for solving the evaluated CPs, e.g., LNS?
>
> __Response:__ Thank you for your question. Regarding this, we did consider comparing SGE with fast suboptimal algorithms, such as Large Neighborhood Search. To address this, we added two experiments to evaluate the performance of SGE against other well performing methods on larger problem sizes than considered in the paper. Table 1 (rebuttal) presents results on Job Shop Scheduling problems with 50 and 100 jobs, while Table 2 (rebuttal) shows experiments on the Vehicle Routing Problem with 100, 150, and 200 nodes. The results analysis shows that SGE performs better than LNS but falls short compared to LKH3 and Google OR-Tools, which are specifically tailored for combinatorial tasks. However, SGE remains applicable to a broader range of tasks (e.g. reasoning tasks). We believe that SGE's performance could improve further once libraries like LKH3 and Google OR-Tools are integrated into GPT-4's code interpreter, allowing our algorithm to leverage these tools within its solution trajectories.
>
>
> __Comment:__ Why is the Integrate step done by the LLMs and not directly by parsing and analyzing the answer so far?
>
> __Response:__ Thank you for your question. Regarding this comment, the Integrate step is handled by the LLMs because the task involves more than simply choosing one of the solution trajectories. Instead, the LLM may combine elements from different solution trajectories to potentially create a more optimized overall solution. By allowing the model to perform this integration, it can either select the best individual solution or combine parts from multiple solutions to enhance performance. We hope this explanation clarifies the rationale behind our approach.

---

> > ### Comment · Reviewer_auHA · 2024-08-12
> > **Thanks! many aspects have been clarified**
> >
> > I am pretty satisfied with the authors' responses.
> > The comparisons to suboptimal solvers is especially exciting to me, and I think it must be in the paper.

---

> > > ### Author Response · Authors · 2024-08-12
> > >
> > > Thank you very much for your feedback. We appreciate your review that have helped us enhance our work.

---

### Official Review · Reviewer_zLRa · 2024-07-13

**Soundness:** 3
**Presentation:** 3
**Contribution:** 3
**Rating:** 8
**Confidence:** 4

**Summary:**

This paper discusses "Self-Guiding Exploration" (SGE), a new prompting strategy designed to enhance the problem-solving capabilities of Large Language Models (LLMs) in addressing Combinatorial Problems (CPs). The authors demonstrate that SGE leverages the autonomy of LLMs to generate and refine solution paths, significantly improving the models' efficiency in dealing with NP-hard problems prevalent in logistics and resource management. SGE operates by generating multiple thought trajectories for each CP task, breaking these into manageable subtasks, and refining the outputs to optimize results. The proposed strategy surpasses existing methods, enhancing CP optimization performance by 27.84% and achieving a 2.46% higher accuracy in various reasoning tasks compared to the best existing results. This research marks a pioneering effort in applying LLMs comprehensively to a range of complex CPs and establishes new benchmarks in both performance and versatility.

**Strengths:**

1. The Self-Guiding Exploration (SGE) strategy is a novel approach that significantly deviates from traditional LLM prompting methods.
2. The paper introduces a new paradigm for using AI for complex problem-solving, enabling the model to generate and refine solution paths autonomously.
3. The experiments are designed and described clearly, making the results seem replicable and understandable. The authors provide extensive empirical results demonstrating the advantages of the SGE approach over existing methods.
4. Applying this new strategy to NP-hard combinatorial problems, critical in many industrial and logistical contexts, represents a substantial advancement.
5. The method's ability to improve optimization performance by over 27% and increase accuracy in reasoning tasks by 2.46% compared to existing best results is statistically and practically significant.
6. The SGE method has potential beyond the tested combinatorial problems, given its wide adaptability over different combinatorial problems.

**Weaknesses:**

1. Performance improvements are reported primarily in LLMs. The paper should address how the strategy scales with compact models, which are more accessible for practical applications and deployment.
2. The paper could strengthen its argument by providing a broader comparative analysis with other state-of-the-art methods, especially those using different AI approaches to solve combinatorial problems (such as approximation algorithms or heuristics). This would help validate the superiority of SGE across a wider range.
3. The paper occasionally glosses over deep technical details and assumptions within the SGE framework. More detailed explanations of the underlying mechanisms, particularly how the subtasks are autonomously generated and refined, would enhance the paper's transparency and reproducibility.
4. The computational cost of running multiple explorations and refinements on LLMs is not addressed. A detailed cost-benefit analysis would be pertinent, especially for potential adopters who are weighing the economic and environmental implications of such advanced AI techniques.

**Questions:**

1. Can the authors elaborate on how the Self-Guiding Exploration strategy scales with varying sizes of LLMs and combinatorial problems? Specifically, how does SGE perform under constraints of lower computational resources or with compact models?
2. How robust is the SGE strategy when applied to combinatorial problems outside the tested domains, such as logistics and scheduling? Are there specific types of CPs where SGE might not perform as well?
3. While the focus has been on NP-hard problems, has there been any exploration of the applicability of SGE to problems that do not fall into this category? What modifications, if any, would be necessary to adapt SGE to such contexts?
4. Could the authors compare SGE with other AI methods (such as approximation algorithms or heuristics) currently considered state-of-the-art in solving similar combinatorial problems?

**Limitations:**

1. The strategy's effectiveness relies heavily on large language models' capacities and specific configurations. This dependency could limit the applicability in environments where such models are not feasible due to resource constraints or accessibility issues.
2. The paper primarily presents results for specific problem sizes. It would be beneficial to include an analysis of how well the SGE strategy generalizes across a broader range of problem sizes, particularly large state spaces common in real-world applications.
3. While the paper provides quantitative improvements, the experimental setup could be expanded to include more diverse datasets and problem scenarios to validate the SGE method's robustness and consistency.
4. A significant limitation is the lack of a detailed analysis of the computational, environmental, and operational costs of implementing SGE.
5. The paper does not discuss how the SGE strategy adapts to evolving data or dynamic environments, which is critical for applications in rapidly changing fields such as logistics and real-time scheduling.

---

> ### Author Rebuttal · Authors · 2024-08-05
>
> __Comment:__ The computational cost of running multiple explorations and refinements on LLMs is not addressed.
>
> __Response:__ Thank you for bringing this important aspect to our attention. Regarding this comment, we have added a new analysis to address the computational cost of running multiple explorations and refinements on large language models.
>
> To evaluate the cost-effectiveness of our approach, we conducted experiments on the Vehicle Routing Problem using the GPT-4 model. Table 3 (rebuttal) presents the average computational cost associated with these experiments. However, while the costs are substantial for large models like GPT-4, using open-source models such as LLaMA can significantly reduce expenses, particularly when running on personal hardware.
>
> We have incorporated this new analysis into the main paper to strengthen the discussion on the economic implications of our approach. We hope this addresses your concern.
>
> __Comment:__ Can the authors elaborate on how the Self-Guiding Exploration strategy scales with varying sizes of LLMs and combinatorial problems?
>
> __Response:__ Thank you for your question. Regarding this comment, we acknowledge the importance of understanding how the SGE strategy scales with varying sizes of LLMs and different combinatorial problems.
>
> To clarify, the performance of SGE does improve with larger models; however, the scale of improvement is not linear. For instance, while the LLAMA-7B model exhibits lower performance compared to the LLAMA-70B model, the difference in performance is not proportional to the difference in model size. This suggests that while larger models offer better results, smaller models like LLAMA-7B still perform reasonably well, though not as effectively.
>
> We did not include results from smaller LLMs in our paper because open-source models smaller than LLAMA-7B generally perform worse. However, we believe that fine-tuning these smaller models using supervised learning with examples of methods to solve combinatorial problem tasks could potentially enhance their performance.
>
> Regarding larger models, as presented in Tables 1 and 2 (rebuttal), the results show that SGE performs comparably to other solvers even with larger problem sizes.
>
> __Comment:__ Are there specific types of CPs where SGE might not perform as well?
>
> __Response:__ Thank you for this thoughtful question. Regarding this comment, we acknowledge the importance of understanding the robustness of the SGE strategy across different combinatorial problem domains.
>
> In our current study, SGE has shown performance improvements in the tested domains (Table 1, paper), including Vehicle Routing and Job Shop Scheduling problems. However, we recognize that real-world applications often involve more complex scenarios, such as stochastic or dynamic versions of these problems.
>
> While testing SGE on stochastic and dynamic combinatorial problems is indeed a valuable direction for future research, it is a distinct area within combinatorial optimization and is beyond the scope of our current work. We plan to explore this in subsequent studies.
>
> __Comment:__ What modifications, if any, would be necessary to adapt SGE to such contexts?
>
> __Response:__ Thank you for your question. Regarding this comment, our primary goal in designing the SGE method was to develop a general approach that could be applied across a broad range of problem types, such as both combinatorial and reasoning tasks.
>
> To validate the robustness of our model, we extended our experiments to include reasoning tasks that are well-established benchmarks in the LLM research community. The results of these tests are presented in Table 3 of the paper.
>
> To achieve this, we have employed the same algorithm without modifications across these contexts. This design choice was intended to ensure that SGE remains versatile and effective across various problem domains.
>
> We acknowledge that specific modifications to SGE could potentially improve its performance for certain types of problems that do not fall into the NP-hard category. However, our main objective was to create an algorithm that balances generality with strong performance across a wide spectrum of problems, and we believe the current implementation of SGE achieves this balance.
>
> __Comment:__ Could the authors compare SGE with other AI methods?
>
> __Response:__ Thank you for your insightful comment. Regarding this comment, we already provided a comparison between SGE and Google OR-Tools in our original submission, as shown in Table 2. In these experiments, we focused on small-size problems where globally optimal solutions could be found using a full search via Google OR-Tools, employing depth-first search to exhaustively explore all possible solutions.
>
> However, to further clarify your comment and expand the breadth of our analysis, we have conducted additional experiments and included new comparative results. To evaluate the performance of SGE against other state-of-the-art methods, we conducted experiments on larger problem sizes against various combinatorial solvers. Table 1 (rebuttal) presents results on Job Shop Scheduling problems with 50 and 100 jobs, while Table 2 (rebuttal) shows experiments on the Vehicle Routing Problem with 100, 150, and 200 nodes. The results analysis shows that SGE performs better than LNS but falls short compared to LKH3 and Google OR-Tools, which are specifically tailored for combinatorial tasks. However, SGE remains applicable to a broader range of tasks (e.g. reasoning tasks). We believe that SGE's performance could improve further once libraries like LKH3 and Google OR-Tools are integrated into GPT-4's code interpreter, allowing our algorithm to leverage these tools within its solution trajectories.
>
> We have incorporated these new tables into the main paper to provide a more comprehensive analysis. We hope this addresses your concern.

---

> ### Author Response · Authors · 2024-08-05
> **One limitation response that did not fit into the rebuttal**
>
> __Comment:__ The paper should address how the strategy scales with compact models.
>
> __Response:__ Thank you for highlighting this important consideration. Regarding this comment, we already evaluated the scalability of our strategy with more compact models in our paper. Specifically, we included experiments with the LLaMA models, as detailed in Figure 4 (paper).
>
> To clarify your comment further, we presented results for both the LLaMA-2-70B model, which requires around 140 GB of VRAM, and the LLaMA-2-7B model, which operates on just 14 GB of VRAM. Additionally, the LLaMA-2-7B model, when quantized, achieves a 4x reduction in VRAM requirements, allowing it to run on a single NVIDIA RTX GPU. This demonstrates the model's suitability for both the research community and practical deployments.
>
> We hope this answers your concern regarding the scalability of our strategy with compact models. However, if you are referring to even smaller models, as mentioned in the paper and noted in the limitations section, SGE heavily depends on the underlying LLM, and smaller models like 1B do not exhibit strong performance. Once again, we appreciate your attention to this aspect of our work.

---

> > ### Comment · Reviewer_zLRa · 2024-08-08
> >
> > Thank you for your response and additional experiments. All my concerns are addressed and I have updated my score.

---

> ### Author Response · Authors · 2024-08-08
>
> Thank you once again for your valuable feedback.

---

### Author Rebuttal · Authors · 2024-08-05

__Response to All:__ We would like to extend our sincere thanks to all the reviewers for their valuable work and insightful comments. We carefully considered the feedback provided and made significant improvements to our paper based on your suggestions.

__Addressing Reviewer Concerns:__ One of the primary concerns shared by the reviewers was the lack of experiments with well-known solvers and heuristics designed for combinatorial problems, as well as the need for testing on larger CP instances. In response, we conducted additional experiments and compiled the results into two new tables, which include comparisons of our method against other solvers and heuristics on larger instances of Job Shop Scheduling and Vehicle Routing Problems (Tables 1 and 2 below). These tables have been incorporated into the revised paper. During our own review, we also identified some errors in the large tables in the appendix and in Table 2 of the paper. Specifically, while conducting experiments versus the global optimum found using the depth-first search method in Google OR-Tools, we mistakenly filled in the results from the LLaMA-2-70B model instead of GPT-4. We will correct this in the final version of the paper. Additionally, we have included new literature and a new baseline that utilizes large language models to create heuristics for CPs (Table 4 below). Recognizing the importance of understanding computational costs, we also added a table detailing the cost of running our algorithm (Table 3 below).

__Clarifying Our Objective:__ We want to clarify that our primary objective was to enhance existing prompting strategies, aiming to make them both generalizable and effective across different tasks. We initially chose combinatorial problems due to their inherent challenges and then extended our approach to demonstrate its generalizability on reasoning tasks. It is important to note that our goal was not to surpass state-of-the-art algorithms in CP research, as these are typically fine-tuned for specific tasks (like neurosolvers do not compete with exact methods). However, we believe that with the continued development of LLMs, they will eventually be able to incorporate both neurosolvers and exact methods in their solution trajectories.

__Research Focus:__ Our exploration focused on several approaches in large language model research: the structure of thought, decomposition, and refinement methods. These approaches are general and effective on simple reasoning tasks, so we aimed to explore whether a general algorithm could also work well on more complex tasks, such as CPs without task-specific modifications.

__Conclusion:__ Once again, we express our gratitude to all the reviewers and area chairs for their efforts and the organization of the review process. The modifications we made have greatly improved the quality of our paper.


__Table 1.__ Percentage performance improvement compared to IO prompting on Job Scheduling Problem. Columns show the number of n jobs and m machines.
| | n50m10 | n50m20 | n100m10 | n100m20 |
| --- | --- | --- | --- | --- |
| LNS | 57.2 | 59.1 | 59.6 | 60.8 |
| OR-Tools | 61.3 | 63.1 | 62.4 | 61.7 |
| SGE | 59.1 | 62.9 | 61.4 | 60.8 |


__Table 2.__ Percentage performance improvement compared to IO prompting on Vehicle Routing Problem. Columns show the number of nodes.
| | n100 | n150 | n200 |
| --- | --- | --- | --- |
| LNS | 57.8 | 58.7 | 58.1 |
| OR-Tools | 62.5 | 61.2 | 60.3 |
| LKH3 | 65.3 | 64.4 | 65.8 |
| SGE | 59.6 | 60.1 | 59.8 |


__Table 3.__ VRP average total cost.
| Number of Nodes | Total Cost |
| -------- | ------- |
| 5 | $0.0961 |
| 8 | $0.1676 |
| 12 | $0.1964 |
| 20 | $0.3515 |


__Table 4.__ Percentage performance improvement compared to IO prompting on Job Scheduling Problem. Columns show the number of nodes.
| | n50m10 | n50m20 | n100m10 | n100m20 |
| --- | --- | --- | --- | --- |
| EoH | 57.8 | 59.6 | 56.4 | 57.1 |
| SGE | 59.1 | 62.9 | 61.4 | 60.8 |

---

### Decision · Program_Chairs · 2024-09-25

**Decision:**

Accept (poster)

**Comment:**

The paper proposes an LLM prompting algorithm for solving combinatorial optimization problems (COP). Some reviewers had concerns about the importance of the problem. However, studying this subclass of reasoning problems turns out to be interesting. Among the weakness, the reviewers pointed out the lack of comparison with COP tools and not reporting inference time. The rebuttal answered these points, offering comparison with the direct approach of just asking the solution (dubbed IO in the paper). The consensus is that the problem and the algorithm are interesting. I agree with the consensus.